# Neutralizing the pathological effects of extracellular histones with small polyanions

Connor H. O' Meara [1,8], Lucy A. Coupland [1,8], Farzaneh Kordbacheh[1,8], Benjamin J. C. Quah[1,8], Chih-Wei Chang [2,8], David A. Simon Davis[1], Anna Bezos[1], Anna M. Browne [1], Craig Freeman [1], Dillon J. Hammill[1], Pradeep Chopra[2], Gergely Pipa[2], Paul D. Madge[2], Esther Gallant[3], Courtney Segovis[3], Angela F. Dulhunty[3], Leonard F. Arnolda[4], Imogen Mitchell[5], Levon M. Khachigian [6], Ross W. Stephens [7], Mark von Itzstein [2] & Christopher R. Parish [1✉]

Extracellular histones in neutrophil extracellular traps (NETs) or in chromatin from injured tissues are highly pathological, particularly when liberated by DNases. We report the development of small polyanions (SPAs) (~0.9–1.4 kDa) that interact electrostatically with histones, neutralizing their pathological effects. In vitro, SPAs inhibited the cytotoxic, platelet-activating and erythrocyte-damaging effects of histones, mechanistic studies revealing that SPAs block disruption of lipid-bilayers by histones. In vivo, SPAs significantly inhibited sepsis, deep-vein thrombosis, and cardiac and tissue-flap models of ischemia-reperfusion injury (IRI), but appeared to differ in their capacity to neutralize NET-bound versus free histones. Analysis of sera from sepsis and cardiac IRI patients supported these differential findings. Further investigations revealed this effect was likely due to the ability of certain SPAs to displace histones from NETs, thus destabilising the structure. Finally, based on our work, a non-toxic SPA that inhibits both NET-bound and free histone mediated pathologies was identified for clinical development.

[1] ACRF Department of Cancer Biology and Therapeutics, The John Curtin School of Medical Research, The Australian National University, Canberra, ACT 2601, Australia. [2] Institute for Glycomics, Griffith University, Gold Coast, QLD 4222, Australia. [3] Eccles Institute of Neuroscience, John Curtin School of Medical Research, Australian National University, Canberra, ACT 2601, Australia. [4] Illawarra Health and Medical Research Institute, Wollongong, NSW 2500, Australia. [5] Intensive Care Unit, The Canberra Hospital, Garran, Canberra, ACT 2605, Australia. [6] Vascular Biology and Translational Research, School of Medical Sciences, University of New South Wales, Sydney, NSW 2052, Australia. [7] Department of Applied Mathematics, Research School of Physics and Engineering, The Australian National University, Canberra, ACT 2601, Australia. [8] These authors contributed equally: Connor H. O'Meara, Lucy A. Coupland, Farzaneh Kordbacheh, Benjamin J. C. Quah, Chih-Wei Chang. ✉email: christopher.parish@anu.edu.au

Histones package DNA into nucleosomes within cell nuclei and regulate gene expression via their N-terminal tails[1]. An additional totally different role of histones occurs in innate immunity where neutrophils, following activation by pathogens, platelet signaling, or extensive tissue injury, extrude chromatin as complex extracellular networks, called neutrophil extracellular traps (NETs), that capture and kill bacteria[2,3]. NETs carry a range of granule-derived and nuclear proteins, including the antimicrobial enzymes myeloperoxidase and neutrophil elastase, but histones predominate, constituting ~70% of NET-associated proteins[4]. NETs are degraded by both endogenous and microbial-derived nucleases, hence histones may be found within the bloodstream associated with DNA fragments as nucleosome-containing structures or as DNA-free histones.

Due to their highly cationic nature, DNA-free histones are cytotoxic not only for microbes but also for host cells[5–7]. DNA-free histones attack glomerular basement membranes[8], initiate coagulation by activating platelets[9,10], and bind erythrocytes, inducing phosphatidylserine exposure and enhancing erythrocyte fragility and rigidity resulting in splenic retention and anemia[11,12]. These effects lead to the development of micro-thrombi and extensive tissue and organ damage resembling the pathology seen in sepsis. In fact, evidence suggests that extra-cellular histones may play a role in organ failure and death in sepsis and other critical illnesses[6,7,13–15]. Damaged tissues also release large amounts of chromatin and associated histones into the circulation that induce sepsis-like symptoms and death as occurs in severe trauma[6,16,17]. NETs and their histones are also major mediators of several vascular pathologies, such as athero-sclerosis[18–21], ischemia–reperfusion injury (IRI)[22,23], deep-vein thrombosis (DVT)[24,25] and stroke[26]. In addition, a recent study in COVID-19 patients correlated serum NET levels with the severity of respiratory disease[27]. Furthermore, it appears that NETs and associated histones play a role in autoimmune dis-eases[28–30] and even in the development of gallstones[31]. Thus, NETs are referred to as a two-edged sword, controlling infections and resolving inflammation, but at the same time having pro-found pathological effects[32–35]. Based on these findings, there is a need for drugs that neutralize the damaging effects of extra-cellular histones, either DNA free or NET associated, without impeding the beneficial effects of neutrophils and NETs. Such drugs must be well tolerated at high doses as circulating histones have been measured at almost mg ml$^{-1}$ concentrations in some trauma patients[6].

In this study, we undertook a drug discovery program based on the hypothesis that, since histones are highly cationic, small polyanionic molecules that generally have a favorable safety profile, should interact electrostatically with free histones and NETs and neutralize their pathological effects.

## Results

### SPAs prevent histone-mediated damage of endothelial cells and RBC.

It is well known that polycations, such as histones, are highly toxic for cells[36] and that the anionic polysaccharides heparin[37–39] and polysialic acid[40–42] very efficiently bind to and neutralize the cytotoxic activity of histones, and in the case of heparin, it has an effect independent of heparin's anticoagulant activity[37–39,43]. Similarly, we have reported that in vivo circulat-ing histones bind to heparan sulfate (HS) in the lung vasculature, an interaction blocked by heparin[44]. These studies indicate that saccharides represent an excellent chemical backbone for the synthesis of O-sulfated small polyanions (SPAs) that may inhibit histone cytotoxicity, with sulfate the preferred anionic group as it is a much stronger anion than carboxyl or phosphate groups. Using this approach, we searched for SPAs with well-defined

structures exhibiting histone-neutralizing activity comparable to heparin, but, due to their small size and structural homogeneity, lack the many and clinically significant off-target effects of heparinoids[45].

To quantify the cytotoxicity of histones for different cell lines and the ability of different polyanions to neutralize this cytotoxicity, we used a flow cytometry-based assay. In this assay, dead cells are detected by propidium iodide (PI) uptake and viable cells by calcein-AM retention (Supplementary Fig. 1a), with histones being cytotoxic for human microvascular endothe-lial cell-1 (HMEC-1) and human umbilical vein endothelial cells (HUVECs) in a concentration-dependent manner (Supplemen-tary Fig. 1b). We then assessed a total of 46 different compounds, consisting of 15 glycosaminoglycans (GAGs) and 29 SPAs, of which 15 were synthesized for this project (see Supplementary information—Chemistry and Supplementary Fig. 12) and, as controls, 2 unsulfated saccharides for their ability to protect HMEC-1 from histone cytotoxicity, the data being presented as half-maximal inhibitory concentration (IC$_{50}$) values (Fig. 1a, left panel). Many GAGs effectively inhibited histone cytotoxicity, with the exceptions being hyaluronic acid, keratan sulfate, and, to a lesser extent, under-sulfated HS (HS$^{lo}$). The inhibitory activity of heparin was retained in low molecular weight (LMW)-heparin (enoxaparin) and heparin with single moieties (i.e., COO$^-$, O- or N-sulfate groups) chemically removed, but globally O- and N-desulfated heparin was inactive. The differently modified cyclodextrins were also informative, the sulfated form being highly inhibitory but the carboxylated and phosphorylated derivatives inactive, consistent with sulfated saccharides being the most potent inhibitors of histone-mediated cytotoxicity.

However, the most remarkable finding, revealed in Fig. 1a, is that SPAs as small as sulfated di- and tri-saccharides of hexoses are as effective as unfractionated heparin at neutralizing histone-mediated cytotoxicity, a decline in histone-inhibitory activity only being observed with sulfated monosaccharides. Such a result is surprising as previous studies with other heparin-binding proteins (e.g., enzymes, growth factors, che-mokines, and coagulation factors) show that a sulfated pentasaccharide-like structure is required for substantial bind-ing, there being negligible binding of such proteins to sulfated di- and tri-saccharides[46–48]. It is noteworthy, however, that the pentose-based disaccharide, xylobiose per-O-sulfate, was less effective at inhibiting histone cytotoxicity, and its methylated version, methyl β-xylobioside per-O-sulfate was inactive (Fig. 1a, left panel). This difference is presumably due to the lower level of sulfation of pentose- versus hexose-based saccharides. In fact, the hexose-based disaccharide, methyl cellobioside per-O-sulfate, whether methylated as the α- or β-anomer, remained a potent inhibitor of histone-mediated cytotoxicity. Collectively, these data indicate that, on a weight basis, a highly sulfated hexose disaccharide is the minimum structure required for a SPA to inhibit histone-mediated cytotoxicity as effectively as heparin. All these data were obtained with HMEC-1 in tissue culture medium containing 10% fetal calf serum (FCS), but very similar results were generated when more physiologically relevant concentrations of FCS were used, namely, 40% serum (Supplementary Fig. 2), which resembles the protein content of interstitial fluids that many cells encounter[49].

We reported recently that histones induce erythrocyte fragility[12] and thus the same panel of 46 compounds was tested for inhibition of histone-mediated erythrocyte fragility (Fig. 1a, right panel). A pattern of inhibition was observed that was almost identical to that seen with histone-mediated HMEC-1 cytotoxi-city, the only substantive difference being cyclodextrin phosphate that partially inhibited histone-mediated erythrocyte fragility, but did not inhibit histone cytotoxicity.

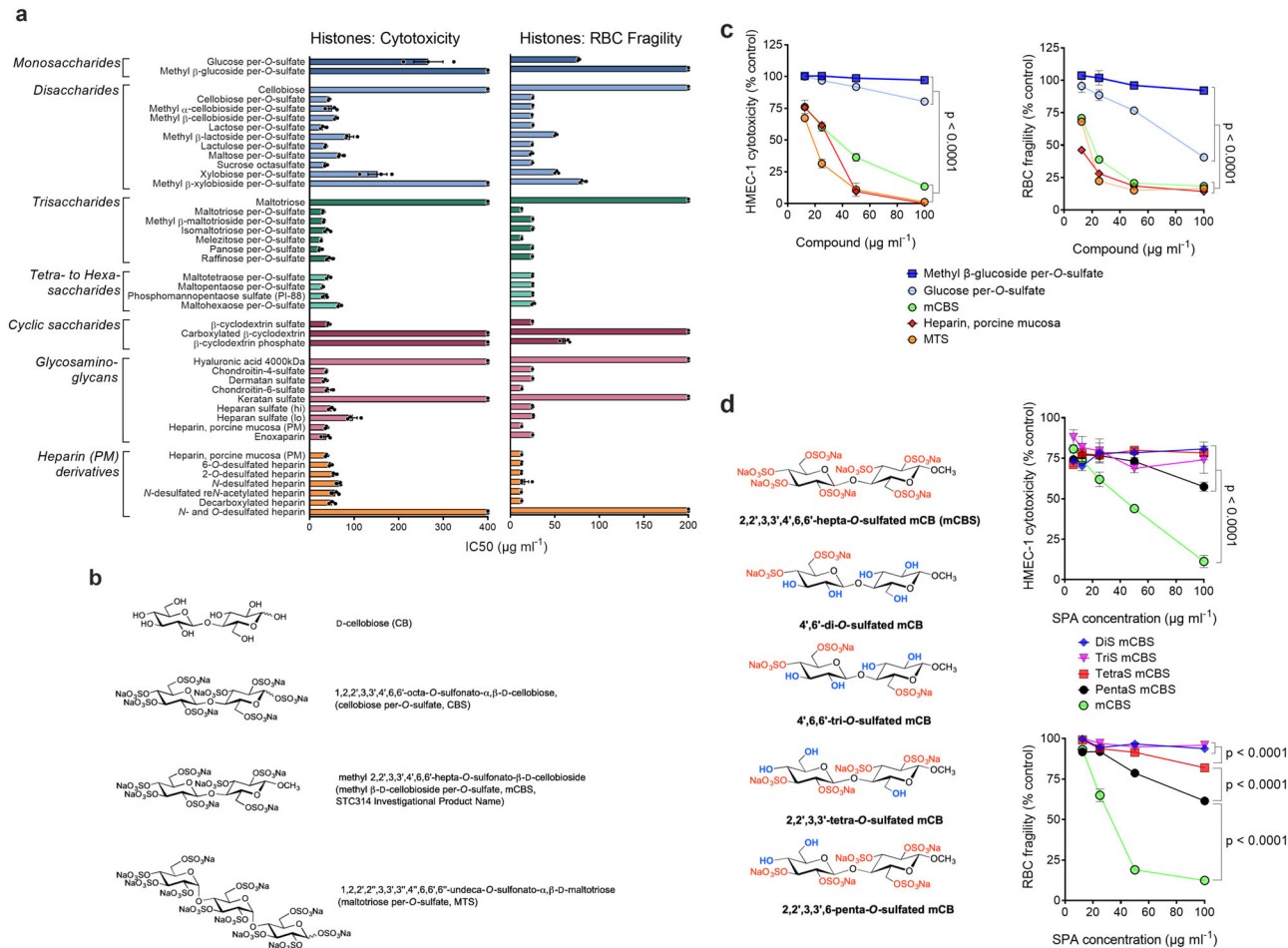

**Fig. 1 Polyanions inhibit histone-mediated endothelial cell cytotoxicity and RBC fragility with minimal structural requirements for the activity identified. a** Human endothelial cells (HMEC-1) (left panel) or red blood cells (RBCs) (right panel) were incubated (1 h, 37 °C) with histones (400 μg ml$^{-1}$) in the presence of different polyanion concentrations and IC$_{50}$ values (mean of three separate determinations ± s.e.m.) for HMEC-1 cytotoxicity or RBC fragility determined. Heparan sulfate (lo) and (hi) represent low and highly sulfated heparan sulfate. **b** Chemical structures of compounds selected for future study. **c** Sulfated disaccharide mCBS has minimal structural requirements to approximate inhibitory effect of heparin on histone-mediated cytotoxicity and RBC fragility, the monosaccharides, methyl β-glucoside per-*O*-sulfate and glucose per-*O*-sulfate having little or no activity. **d** Effect of the level of sulfation of mCB (di-, tri-, tetra-, and penta-*O*-sulfated) on its ability to be an effective inhibitor of histone-mediated cytotoxicity and RBC fragility compared to fully sulfated (hepta-*O*-sulfated) mCBS. Data are presented as mean ± s.e.m. (*n* = 3) and analyzed by two-way ANOVA with Tukey's correction for multiple comparisons. Source data are provided as a Source Data File.

We next selected three SPAs for future study, based on our findings in Fig. 1a, namely cellobiose per-*O*-sulfate (CBS), methyl β-cellobioside per-*O*-sulfate (mCBS), and maltotriose per-*O*-sulfate (MTS) (structures in Fig. 1b). In subsequent studies, CBS and mCBS were often used interchangeably, but mCBS is chemically much more stable than CBS and, consequently, represents a better drug candidate (Supplementary Fig. 3). We chose MTS as it was one of the most inhibitory sulfated trisaccharides tested and is consistently more active against histones than the sulfated disaccharides CBS and mCBS. Most importantly, all three SPAs have histone-inhibitory activity similar to heparin, unlike sulfated monosaccharides that are much less active (Fig. 1c). Furthermore, highly sulfated SPAs are required as under-sulfated mCBS had minimal histone-inhibitory activity, even when five of seven *O*-sulfation sites were occupied (Fig. 1d), with unsulfated cellobiose and maltotriose also being inactive (Fig. 1a).

In addition to inducing erythrocyte fragility, we previously demonstrated that histones promote erythrocyte aggregation and reduce erythrocyte deformability[12]; therefore, we investigated whether mCBS and MTS prevent these processes. Based on flow

cytometry forward (FSC) and side scatter (SSC), histones very efficiently aggregate erythrocytes, with this effect being completely inhibited by mCBS (Fig. 2a), a result confirmed by scanning electron microscopy (Fig. 2b). We also used erythrocyte autofluorescence to quantify the number of erythrocytes present in histone-induced aggregates. At high histone concentrations (400 μg ml$^{-1}$), there were ~20 erythrocytes/aggregate, but mCBS and MTS prevented aggregation in a concentration-dependent manner, with MTS being ~2-fold more effective than mCBS (Fig. 2c). Finally, erythrocytes exposed to histone concentrations that do not induce erythrocyte aggregation (≤50 μg ml$^{-1}$) (Fig. 2d) showed significantly reduced passage through an artificial spleen, this assay measuring erythrocyte deformability/rigidity[50]. The addition of mCBS and MTS, however, totally restored the ability of erythrocytes to passage through an artificial spleen (Fig. 2d), indicating that SPAs, as well as protecting erythrocytes from histone-mediated aggregation and fragility, prevent histone-induced rigidity.

**SPAs inhibit platelet activation by histones.** Histones are known to induce platelet activation[9,38]; thus, to investigate the capacity of

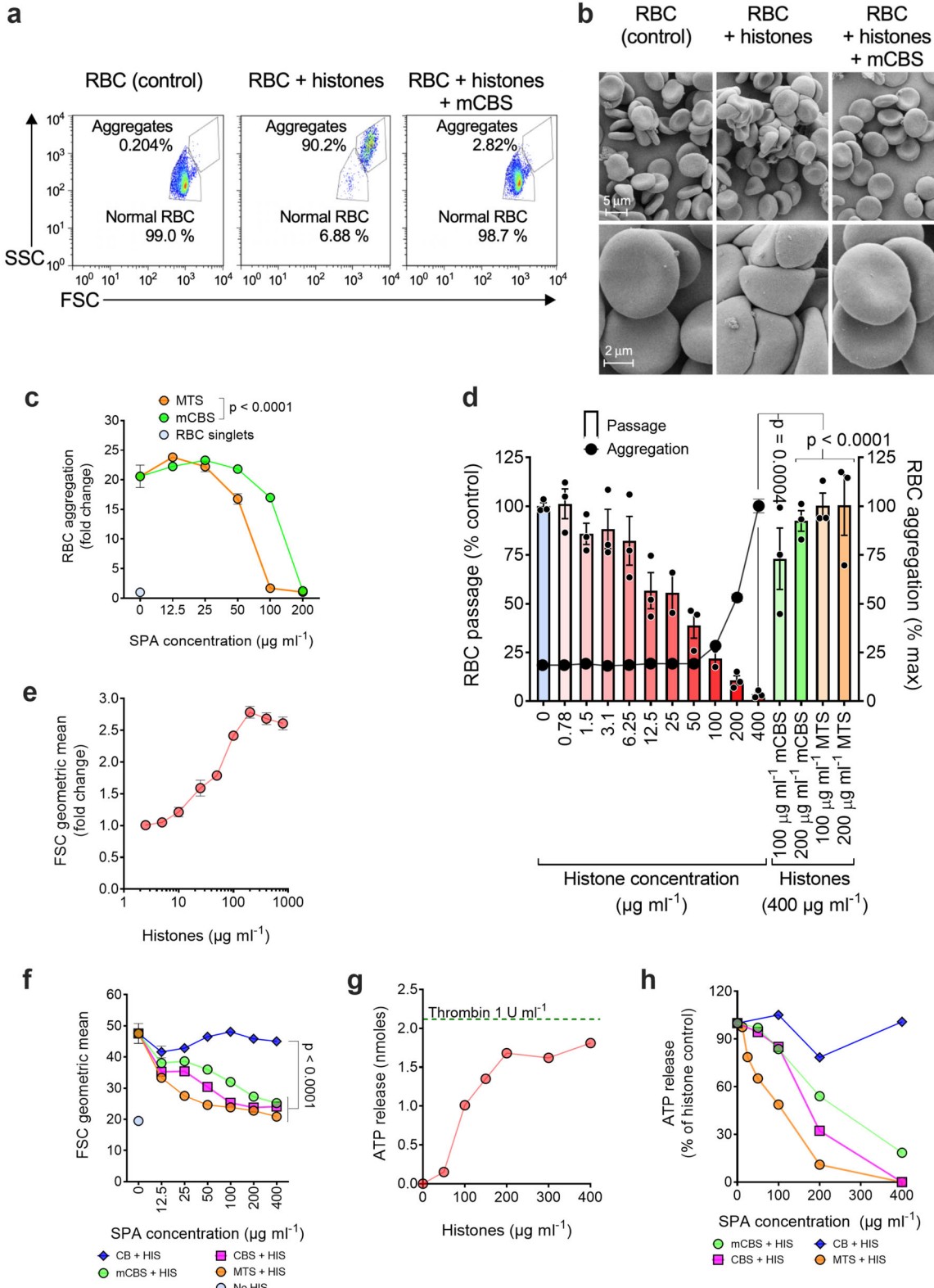

SPAs to inhibit this process, isolated, washed human platelets were incubated with histones and platelet aggregation was measured by flow cytometry. In addition, ATP release, due to platelet degranulation, was measured following exposure of whole blood to histones. Both platelet aggregation and degranulation were histone concentration dependent (Fig. 2e, g), the difference in histone sensitivity of platelets between the two assays being attributed to the presence of plasma proteins and erythrocytes

in the ATP release assay that also bind histones. Histone-induced platelet aggregation was completely inhibited by the three SPAs tested (mCBS, CBS, and MTS), with MTS being the most active (Fig. 2f). Unsulfated CB, included as a negative control, was non-inhibitory (Fig. 2f). Similar results were obtained with histone-induced ATP release (Fig. 2h). Thus, SPAs are effective in inhibiting the platelet-activating properties of histones.

**Fig. 2 Histones promote RBC and platelet aggregation, reduce RBC deformability, and induce platelet degranulation, processes that are inhibited by SPAs. a** Human RBCs were incubated for 1 h at 37 °C alone (control) or in the presence of histones (400 μg ml⁻¹) with or without mCBS (200 μg ml⁻¹). Percentage of RBC aggregation measured by flow cytometry based on forward (FSC) and side (SSC) scatter parameters and an appropriate gating strategy to discriminate aggregated from normal (non-aggregated) RBC. **b** Scanning electron micrographs depicting the level of RBC aggregation at low and high magnification following the three treatments ($n = 1$) depicted in **a** and each representative of three fields per magnification. **c** Concentration-dependent inhibition of histone-mediated RBC aggregation by mCBS and MTS, in this case RBC aggregation being calculated by flow cytometry as fold change in RBC auto-fluorescence relative to RBC in the absence of histones. The fold change values also provide an estimate of the number of RBC in each aggregate. Data were analyzed by two-way ANOVA with Sidak's correction for multiple comparisons ($n = 3$). **d** Retention of RBC in an artificial spleen that measures RBC deformability. RBCs were incubated with increasing concentrations of histones for 1 h and, at the highest concentration used (400 μg ml⁻¹), also incubated with either mCBS or MTS (100 and 200 μg ml⁻¹), prior to passage through the artificial spleen. Data are presented as mean ± s.e.m. ($n = 2$–3) and analyzed by one-way ANOVA with Dunnett's multiple comparisons tests. **e** Histone-induced aggregation of isolated platelets. **f** SPA inhibition of histone (HIS) (150 μg ml⁻¹) induced platelet aggregation. Data were analyzed by two-way ANOVA with Dunnett's correction for multiple comparisons ($n = 6$). **g** Histone-induced degranulation of platelets in whole blood, as measured by ATP release. Dotted line ATP release from thrombin-activated platelets ($n = 1$). **h** SPA inhibition of histone-induced platelet degranulation. Data in **g**, **h** are representative of one of two experiments. Source data are provided as a Source Data File.

**SPAs prevent lipid bilayer disruption by histones**. We next investigated how histones mediate their cytotoxicity and, consequently, how SPAs protect cells from histone-mediated damage. Since histones bind GAGs, notably HS, which are ubiquitously expressed on cell surfaces, it seemed feasible that histones initiate their cytotoxic effector function by binding to cell surface HS. To test this idea, HMEC-1 cells were depleted of cell surface HS by incubation with either a mixture of three bacterial heparinases or human platelet heparanase (HPSE) prior to exposure to histones, with HS removal being 86% and 97%, respectively, for the two enzymatic treatments as monitored by flow cytometry. We found that pre-treatment of HMEC-1 cells with either bacterial or human HS degrading enzymes had no effect on the sensitivity of the cells to histone-mediated cytotoxicity, the two enzyme pre-treatments also having no effect on HMEC-1 viability (Fig. 3a). To confirm this finding, we used a Chinese hamster ovary (CHO) cell line (pgsA-745) that lacks cell surface GAGs, including HS, due to a mutation in the xylotransferase that initiates GAG chain biosynthesis[51]. Compared to the parent CHO-K1 cell line, loss of cell surface GAGs had little effect on histone cytotoxicity, there being only a small but significant reduction in cytotoxicity at the highest histone concentrations tested (Fig. 3b). Thus, cell surface GAGs are not required for histone-mediated cytotoxicity.

Histones are known to interact with and damage lipid bilayers[52], act as cell-penetrating proteins[53], and induce ion channels based on single whole-cell current recordings[6]. Furthermore, recently in a mouse model of atherosclerosis, the N terminus of histone H4 directly penetrated lipid bilayers and mediated smooth muscle cell damage[54]. Thus, next we investigated whether CBS and MTS could prevent histones from disrupting lipid bilayers. To examine this possibility, initially artificial lipid bilayers were prepared and their susceptibility to histone rupture was detected by changes in current across the bilayers. Lipid bilayers have a finite lifetime, normally ~30 to 120 min[55]. In our experiments, control lipid bilayers containing ryanodine receptor 1 ion channel protein had an average lifetime of 46 ± 4 min, with the addition of histones (1 μM) markedly reducing the lifetime to 5.7 ± 1.2 min (Fig. 3c).

In fact, 13/47 bilayers (28%) broke within 0.3–0.5 min of histone addition, whereas only 2/125 control bilayers (1.6%) ruptured in the same time period, with higher histone concentrations (≥50 μM) resulting in rapid rupture of most bilayers (not shown). Bilayers were less prone to rupture by histones when CBS or MTS was present, the average bilayer lifetime increasing significantly to 18 ± 4 and 36 ± 5 min for CBS and MTS, respectively (Fig. 3c). Furthermore, bilayer lifetimes with MTS were not significantly different from the control lifetimes. Similarly, compared with histones alone (28%), the

incidence of rapid bilayer rupture decreased to 3/52 bilayers (5.8%) for CBS and 1/40 bilayers (2.5%) for MTS.

Earlier studies demonstrated that histones can induce non-selective Ca²⁺ channels in cells and plasma membrane depolarization[6,56]. These findings further support the concept that histones directly interact with cell surface phospholipids and disrupt membrane integrity. To investigate whether SPAs protect cells against histone-induced Ca²⁺ flux, HMEC-1 were loaded with the Ca²⁺-sensitive dye, Indo-1, challenged with histones in the presence or absence of CBS or MTS, and Ca²⁺ uptake measured by flow cytometry (Fig. 3d). Histones induced a >8-fold increase in the population of cells exhibiting high intracellular Ca²⁺ levels, this response plateauing 1–3 min after histone addition. The presence of MTS totally ablated the Ca²⁺ response and CBS substantially inhibited the response (Fig. 3d, lower panel). Collectively, our findings confirmed that histones damage cell membranes by directly disrupting the lipid bilayer of cells, with our selected SPAs neutralizing this undesirable property of histones.

**SPAs inhibit several histone-associated pathologies**. We next assessed the ability of SPAs to inhibit histone-mediated pathologies in vivo. Injection of histones i.v. into mice results in a sepsis-like syndrome involving liver damage, generalized tissue injury, and kidney failure, as measured by circulating alanine aminotransferase (ALT), lactate dehydrogenase (LDH), and creatinine levels. Administration of CBS, mCBS, and MTS 10 min prior to histones protected animals from each of these tissue-specific pathologies in a dose-dependent manner, whereas CB was inactive (Fig. 4a). When heparin was used as a comparator, it had a similar inhibitory activity to mCBS (Fig. 4a), but at doses >6.25 mg kg⁻¹ at least 50% of mice had to be euthanized due to bleeding. Systemic histones also induce thrombocytopenia and anemia that was prevented by mCBS and MTS treatment (Fig. 4b). In the case of anemia, this conclusion is based on a decline in circulating red blood cells (RBCs) and a concomitant accumulation of hemoglobin (Hb) in the spleen. To examine whether SPAs influence the localized vascular effects of histones, a histone-mediated model of DVT[57] was established, which revealed this to be almost totally inhibited by both CBS and MTS (Fig. 4c), consistent with both systemic and localized pathologies mediated by free histones being amenable to inhibition by SPAs.

Additional experiments examined whether mCBS and MTS can inhibit ongoing tissue injury induced by histones. In these experiments, mice were injected intravenously (i.v.) with histones, and 2 h later when multiple organ failure is clearly evident, they received mCBS or MTS intraperitoneally (i.p.) and 30 min later

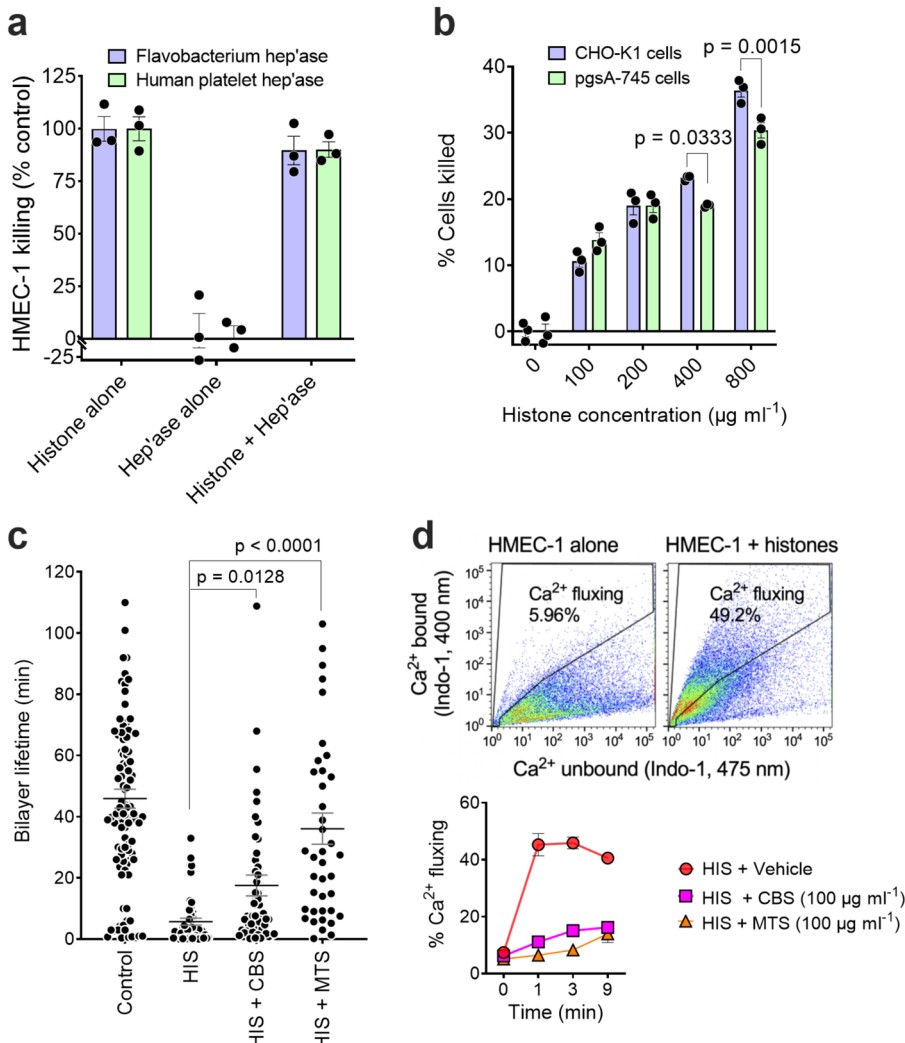

**Fig. 3 Histone-mediated cytotoxicity for cells does not require cell surface heparan sulfate, histones directly disrupting lipid bilayers and inducing a cellular $Ca^{2+}$ flux that can be blocked by SPAs. a** Effect of removal of cell surface heparan sulfate by bacterial heparinases 1, 2, and 3 or human platelet heparanase on the sensitivity of HMEC-1 to histone cytotoxicity. **b** Sensitivity of wild-type CHO-K1 and GAG-deficient pgsA-745 CHO-K1 cells to histone cytotoxicity. Data are presented as mean ± s.e.m. ($n = 3$) and analyzed by two-way ANOVA with Sidak's multiple comparisons test. **c** Lifetime of artificial lipid bilayers exposed to histones (HIS) (1 μM) alone ($n = 47$) or in the presence of the SPAs CBS ($n = 52$) or MTS ($n = 40$) (10 μM). Control bilayers ($n = 125$) contained the RγR1 ion channel protein. Data are presented as mean ± s.e.m. and analyzed by non-parametric Kruskal–Wallis test. **d** Upper panel: Representative flow cytometry plots, using $Ca^{2+}$-sensitive dye Indo-1, showing $Ca^{2+}$ fluxing HMEC-1 1 min following histone addition (100 μg ml$^{-1}$). Lower panel: Time course of effect of CBS and MTS (100 μg ml$^{-1}$) on histone-induced $Ca^{2+}$ flux by HMEC-1. Data are presented as mean ± s.e.m. ($n = 2$) from one of two experiments. Source data are provided as a Source Data File.

were sacrificed, plasma collected, and analyzed for ALT, LDH, creatinine, and Hb content (Fig. 5a). MTS significantly reduced the circulating levels of all four tissue injury markers tested, inhibition in all cases being substantial, that is, 80–90%. In contrast, mCBS inhibitory activity was lower, but with creatinine and Hb levels being significantly reduced.

In parallel studies, we noticed that dead cells could be readily detected by confocal fluorescence microscopy in the organs of histone-treated animals following injection of PI 5 min prior to sacrifice (Fig. 5b). In contrast, few PI+ cells were detected in control animals receiving phosphate-buffered saline (PBS). Thus, this simple assay allowed us to determine whether the SPA mCBS and MTS could inhibit ongoing histone cytotoxicity against liver, lung, and kidney.

We noted that PI+ cells were readily detected in liver, lung, and kidney from histone injected mice, but, based on dead cells per field (Fig. 5c), the prevalence of dead cells was ~10-fold higher

in the kidney compared to the liver and lung. Importantly, the administration of mCBS and MTS 2 h after histone injection resulted in a significant reduction in dead cells in all tissues, with MTS being a more effective inhibitor than mCBS. Thus, these data at the cellular level largely resemble the results obtained at the systemic level and, collectively, demonstrate that SPAs can inhibit multiple organ damage in mice suffering from ongoing histone-induced sepsis.

We next examined the efficacy of SPAs in a rat cecal ligation puncture (CLP) model of sepsis. There was no mortality in the CBS treatment group, an effect that was significant compared with PBS controls, and only one death occurred in the MTS group (Fig. 6a). Importantly, high ALT and creatinine levels detected in the untreated group, indicative of extensive liver and kidney damage, were not seen in the CBS-treated animals, but were only partially reduced in the MTS treatment group (Fig. 6b). This finding is interesting as in vitro and in vivo MTS was

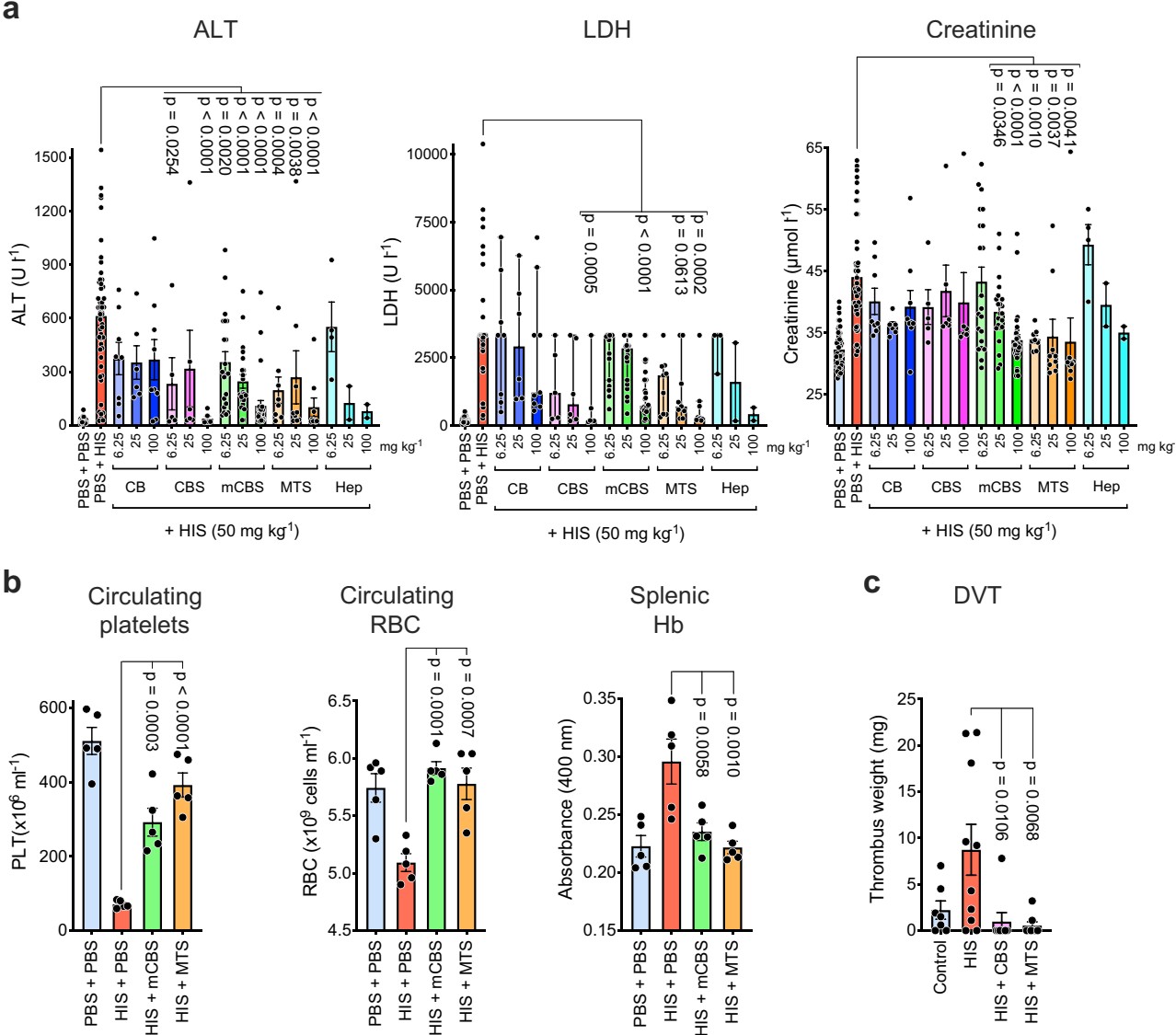

**Fig. 4 In vivo SPAs inhibit histone-induced tissue injury, thrombocytopenia, anemia, and DVT. a** Mice injected i.p. with SPA and heparin (Hep) doses (as indicated) 10 min prior to i.v. injection of histones (50 mg kg$^{-1}$), had blood collected 4 h post histones for assessment of liver (alanine aminotransferase, ALT), kidney (creatinine), and general tissue (lactate dehydrogenase, LDH) damage. Data pooled from 15 separate experiments, with $n = 2$–50 mice per treatment. **b** Mice ($n = 5$/group), treated as above but receiving one SPA dosage (100 mg kg$^{-1}$), had their blood and spleens collected 10 min post histones for assessment of circulating platelets and RBC and splenic hemoglobin (Hb). **c** Impact of CBS and MTS on a mouse model of histone-induced DVT ($n = 7$–10 mice per group). All data, except LDH values, are presented as mean ± s.e.m. and analyzed by one-way ANOVA with Dunnett's correction for multiple comparisons. LDH values at or above the detection limit of 3325 U l$^{-1}$ are reported in the LDH data of panel **a**. For some of these samples, sufficient material was available for dilution to obtain accurate LDH levels. The LDH data are presented as median with 95% confidence intervals, instead of the mean that may be biased by the values at the detection limit. The data above the detection limit was set to 3325 U l$^{-1}$ prior to performing a non-parametric Kruskal–Wallis test with Dunn's correction for multiple comparisons. Source data are provided as a Source Data File.

consistently a more potent neutralizer of DNA-free extracellular histones than mCBS/CBS.

To further investigate this observation, CBS and MTS were tested for their efficacy in a rat cardiac IRI model, previous studies having demonstrated that cardiac IRI is highly NET dependent[22,23]. Remarkably, in this model, MTS treatment had no effect, whereas CBS significantly reduced the area of microvascular obstruction (MVO) and infarct territory in the area at risk (AAR) by 50% (Fig. 6c). This unexpected finding was largely confirmed in a rat skin flap IRI model where mCBS consistently and significantly increased the viable area of the skin flap, whereas the results with MTS treatment were highly variable (Fig. 6d).

**SPAs inhibit NETs in sepsis and acute myocardial infarction patient sera**. Next, we examined sera from sepsis patients for cytotoxic activity as it has been reported that such patients often have high levels of circulating histones and NETs[13–15]. In the presence of 50% patients' serum, our flow cytometry-based assay for HMEC-1 cytotoxicity gave variable results; however, after extensive testing, we found that inhibition of incorporation of $^3$H-thymidine into the DNA of proliferating HMEC-1 gave very reproducible results and served as an excellent indirect measure of histone cytotoxicity and its inhibition by CBS and MTS (Fig. 7a). In fact, sera from sepsis patients inhibited HMEC-1 proliferation, as measured by $^3$H-thymidine incorporation, at a level that correlated with their APACHE II (Acute Physiology

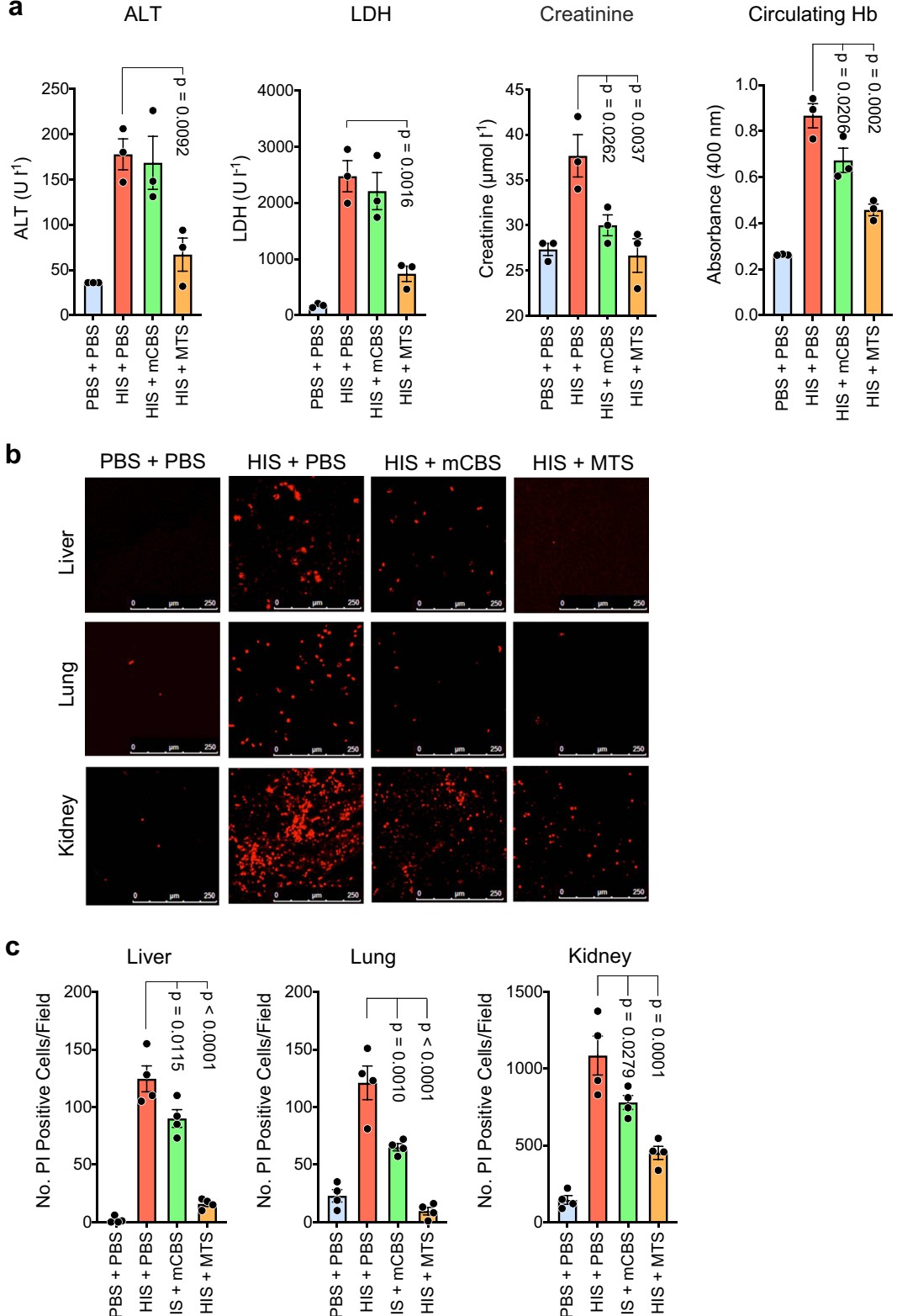

**Fig. 5 In vivo SPAs inhibit ongoing tissue injury induced by histones. a** Mice ($n = 3$/group) were initially injected i.v. with histones (50 mg kg$^{-1}$), and 2 h later, when ongoing tissue injury was evident, they were injected i.p. with mCBS or MTS (100 mg kg$^{-1}$), with plasma collected 30 min later and analyzed for content of ALT, LDH, creatinine, and Hb. **b** Mice were treated as in **a** but 5 min before sacrifice they were injected i.p. with propidium iodide (PI), and liver, lung, and kidneys were collected and examined by confocal fluorescence microscopy for dead (fluorescent red) cells, with representative fields from the various treatments being shown; scale bar 250 μm. **c** Same animals were quantified for the number of dead (PI+) cells per field in the different treatment groups ($n = 4$ mice per group), with at least four fields quantified/organ. Data are presented as mean ± s.e.m. and analyzed by one-way ANOVA with Dunnett's correction for multiple comparisons. Source data are provided as a Source Data File.

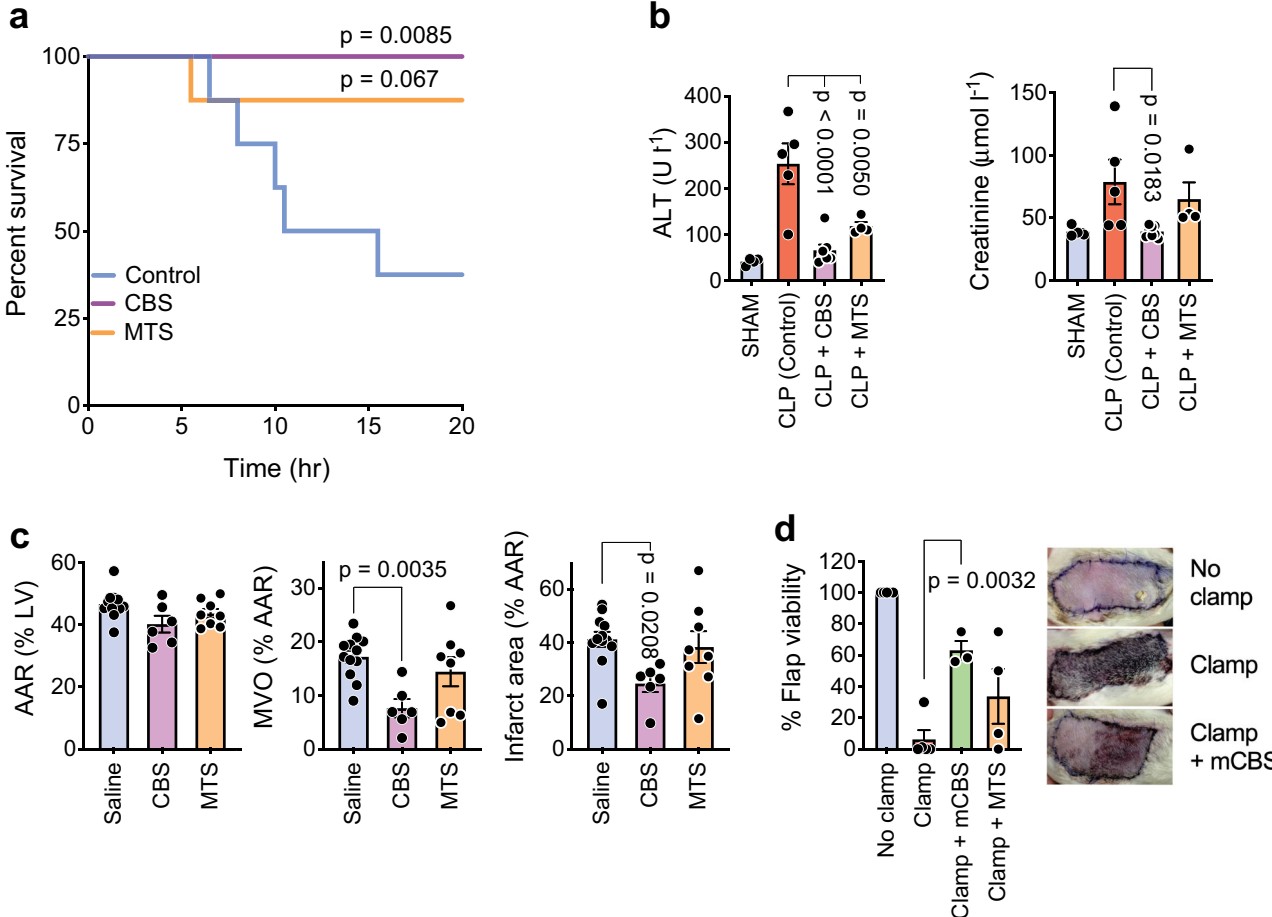

**Fig. 6 SPAs inhibit a range of pathologies involving extracellular histones. a** Survival of rats ($n = 8$ per group) subjected to cecal ligation and puncture (CLP) and receiving saline (Control), CBS, or MTS. *P* values obtained with survival analysis log-rank (Mantel–Cox) test. **b** Kidney and liver damage in CLP rats, as measured by ALT and creatinine blood levels ($n = 4–8$/group, blood could not be collected or was clotted from some rats). **c** Effect of CBS and MTS ($n = 6–12$/group) on cardiac IRI in rats, with the area at risk (AAR) in left ventricle (LV), microvascular obstruction (MVO), and infarct area being measured. **d** Effect of mCBS and MTS on a skin flap model of IRI in rats ($n = 3–5$/group), with representative photos shown (skin flap reduced to 25% normal size). Data are presented as mean ± s.e.m. and analyzed by one-way ANOVA with Dunnett's correction for multiple comparisons. Source data are provided as a Source Data File.

And Chronic Health Evaluation II) scores (Fig. 7b). This correlation was stronger than that observed between APACHE II scores and circulating DNA levels (Fig. 7c). Further analysis of highly inhibitory serum from patient 5 (red circle, Fig. 7b, c) revealed that it had the highest DNA content and its inhibitory activity was DNase I sensitive and also was totally inhibited by polyclonal antibodies specific for histones 3 and 4 (Fig. 7d). Such data are consistent with NETs mediating the anti-proliferative effects of the septic patient's sera. Importantly, we also found that CBS significantly overrode the anti-proliferative activity of the ten most inhibitory septic patient sera, whereas MTS did not (Fig. 7e). This lack of MTS inhibitory activity in vitro resembles that obtained with MTS in vivo in several disease models. In this case, however, it is directly demonstrated that MTS, unlike CBS, fails to overcome NET-mediated inhibition of HMEC-1 proliferation. Thus, it cannot be assumed that SPAs inhibiting pathologies mediated by DNA-free histones are also active against NET-associated histones.

A similar conclusion was made with ST-segment elevation myocardial infarction (STEMI) patients; in this case, sera being collected distal of the coronary occlusion site and immediately after ischemic myocardium reperfusion. STEMI patients were stratified according to their levels of serum troponin I (i.e., ≤35 or >35 µmol l$^{-1}$), with cardiac troponin I

being a cardiac muscle-specific protein that is released into the circulation following myocardial damage, and with circulating troponin I levels correlating with the extent of heart injury[58]. It was found that, compared with control sera, all STEMI patients had elevated levels of extracellular DNA in their occlusion site sera (Fig. 7f), a difference that was highly statistically significant, with higher levels of DNA being observed in patients with high troponin I levels (Fig. 7f). Importantly, CBS partially and significantly reduced the inhibitory activity of the STEMI sera but, in this case, MTS also significantly restored HMEC-1 proliferation, but not to the same extent as CBS (Fig. 7g). These data resemble those obtained with the cardiac IRI model in rats (Fig. 6c), although unlike the rat model, DNA-free histones may play some role in human cardiac IRI.

**Mode of action of SPAs.** So far, our findings indicate that SPAs can effectively inhibit the pathological effects of both DNA-free and NET-associated histones, but SPAs appear to differ in their potency against these two forms of extracellular histones. Thus, mechanistic studies were undertaken to provide a better understanding of the mode of action of SPAs against DNA-free and NET-associated extracellular histones.

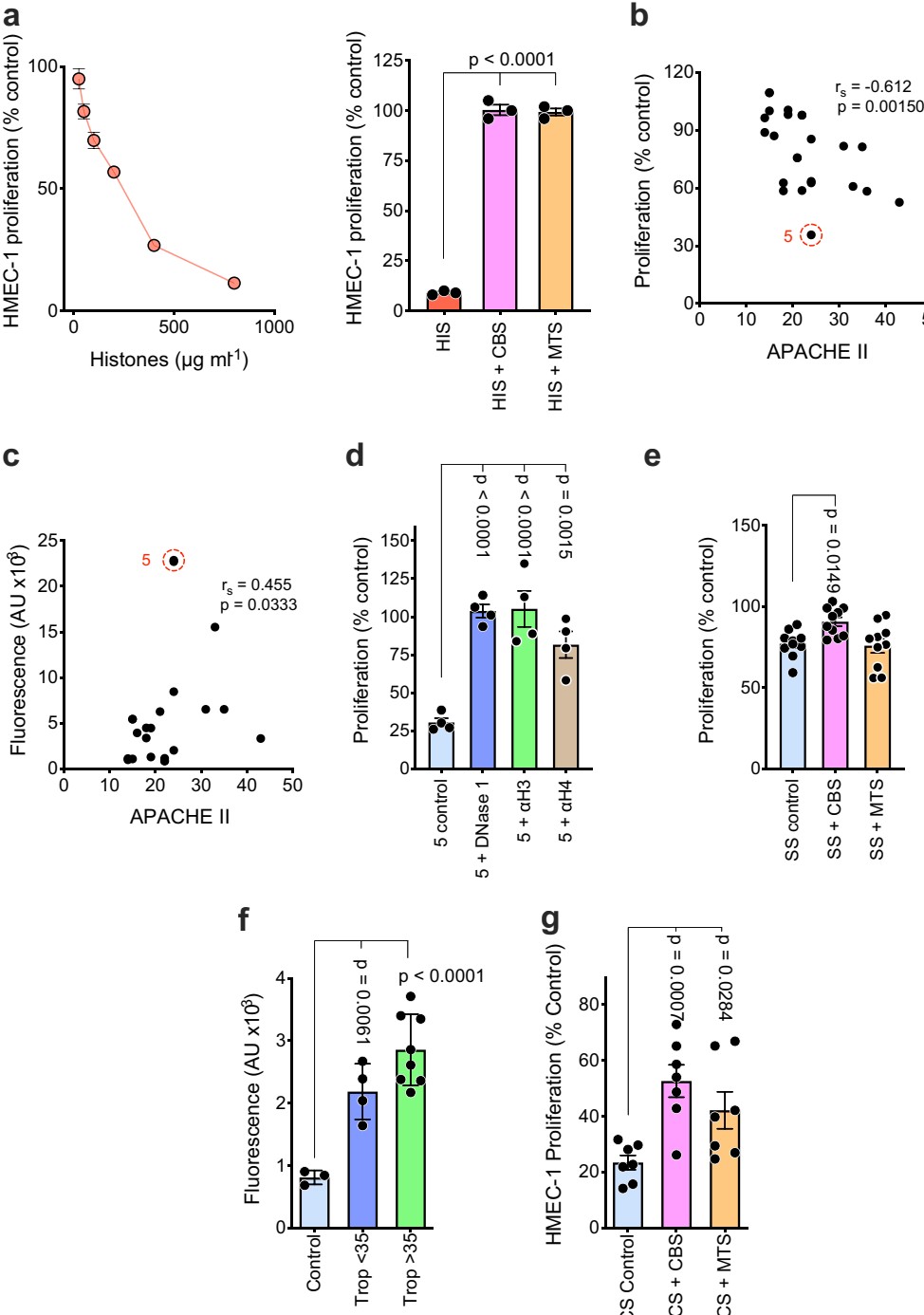

**Fig. 7 Sera from sepsis and acute myocardial infarction patients inhibit endothelial cell proliferation, an effect neutralized by DNase I, anti-histone antibodies, and SPAs. a** Proliferation of HMEC-1, as measured by [3]H-thymidine incorporation, to detect the anti-proliferative activity of histones, which is histone concentration dependent (left panel, $n = 3$), but is totally neutralized by mCBS and MTS (right panel, $n = 3$). **b** Correlation (Spearman's $r$ value) of APACHE II scores with anti-proliferative effect of sepsis patients sera on HMEC-1 ($n = 20$ patients). **c** Correlation of APACHE II scores with extracellular DNA content of sepsis patient's sera, with serum from patient 5 (red circle) having greatest anti-proliferative activity (**b**) and DNA content (**c**). **d** Effect of DNase I or pAbs against histone 3 (αH3 10 μg ml$^{-1}$) and histone 4 (αH4 10 μg ml$^{-1}$) ($n = 4$/treatment) on anti-proliferative effect of serum from sepsis patient 5. **e** Ability of SPAs CBS and MTS to neutralize the anti-proliferative effect of sepsis patient sera (SS) ($n = 10$ patients). **f** Extracellular DNA content of occlusion site sera from STEMI patients ($n = 12$) and stratified according to low (<35 μmol l$^{-1}$, $n = 4$) and high (>35 μmol l$^{-1}$, $n = 8$) troponin I (Trop) levels. Control sera ($n = 3$) from the peripheral blood of healthy volunteers. **g** Ability of SPAs CBS and MTS to neutralize the HMEC-1 anti-proliferative effect of sera from high troponin I STEMI patients (CS) ($n = 7$ patients). Data are presented as mean ± s.e.m. and analyzed by one-way ANOVA with Dunnett's correction for multiple comparisons. Source data are provided as a Source Data File.

**DNA-free histones**. In the course of this study, we noted that the addition of mCBS and MTS to solutions of DNA-free histones resulted in the formation of a precipitate, even in the presence of serum. Experiments were undertaken to quantify the precipitation reaction in vitro and are depicted in Supplementary Fig. 4. Histones (400 μg ml⁻¹) were added to mCBS, MTS, and heparin (400 μg ml⁻¹) and percent precipitation of compounds and histones determined. Relatively high concentrations of the reactants were used because of the low sensitivity of the assays available to detect histones and polyanions in the precipitates. Preliminary studies also indicated that mixing equal concentrations of histones and compounds would result in a substantial proportion (at least 50%) of the histones being precipitated. This was indeed the case, with precipitation of histones by compounds ranging from ~40 to 90% (mCBS < heparin<MTS) and precipitation of compounds by histones from ~60 to 90% (heparin < MTS < mCBS). Thus, SPAs can electrostatically bind to the positively charged surface of histones, precipitate the histones, and mark them for elimination by phagocytes. This process is similar to the anticoagulant activity of heparin being neutralized by the highly cationic peptide, protamine, which is known for decades to precipitate heparin, and the precipitates are cleared by the reticuloendothelial system[59]. Similarly, we discovered that when mCBS and MTS were injected i.p. into mice 10 min prior to i.v. injection of histones and plasma samples collected 0 and 1–2 min later, MTS produced a substantial and significant decline in histone levels (~80%), whereas mCBS had a smaller (~20%) effect (Supplementary Fig. 5a–c). Thus, in vivo, SPAs rapidly (i.e., 1–2 min) eliminate DNA-free histones from the circulation in a process that appears to be a major mechanism of action of these compounds.

**NET-associated histones**. In initial mechanistic studies, we compared the interaction of mCBS and MTS with partially degraded chicken RBC chromatin (Supplementary Fig. 6a); the use of chicken chromatin allows the generation of data not complicated by NET-associated changes in chromatin. Sodium dodecyl sulfate-polyacrylamide gel electrophoresis (SDS-PAGE) was also used to demonstrate that chicken RBC chromatin had a similar content of histone isoforms as calf-thymus histones (Supplementary Fig. 6b). Using this approach, we made the surprising discovery that when chicken RBC chromatin was incubated with MTS or heparin, a concentration-dependent (0.9-500 μg ml⁻¹) increase in fluorescence of the DNA-binding dye Sytox Green was observed, whereas no such increase was seen with mCBS even at 500 μg ml⁻¹ (Fig. 8a). This finding suggested that MTS and heparin displace histones from chromatin, permitting more Sytox Green to bind to the exposed DNA, with mCBS being unable to mediate histone displacement. Previous reports that heparin can displace histones from chromatin and NETs support this interpretation[9,60]. Two-thirds of the chicken chromatin could be pelleted by high-speed centrifugation and SDS-PAGE was then used to determine the histone content of both the pellet and supernatant following exposure of chicken chromatin to PBS, mCBS, MTS, and heparin (Fig. 8b, c). It was found that mCBS (500 μg ml⁻¹) did not change the percentage of histones in the pellet (~65%) from that seen in the PBS control. In contrast, treatment with MTS resulted in only ~35% of the histones appearing in the pellet and with heparin treatment only ~20%, with the majority of histones being in the supernatant. These differences were statistically significant ($p < 0.001$) and confirm that heparin is more effective than MTS at displacing histones from chromatin. Finally, the same experiment was performed with human NETs and generally similar results were obtained despite NET histones being substantially degraded

(Fig. 8d and Supplementary Fig. 6b). Thus, MTS and heparin enhanced Sytox Green binding, but, rather than having no effect, mCBS significantly reduced Sytox Green fluorescence (Fig. 8d), a finding consistently observed with several NET preparations, and possibly due to mCBS binding to histones within the NET structure, thus impacting upon the extent of Sytox Green binding to DNA. Interestingly, with NETs the enhanced Sytox Green uptake induced by MTS and heparin did not titrate out, this difference possibly being due to the ~24-fold lower concentration of histones in the NET versus chicken chromatin preparations (Supplementary Fig. 6b). Collectively, these data suggest that mCBS stabilizes nucleosomes, whereas MTS and heparin promote displacement of histones from nucleosomes and disassembly of chromatin and NETs.

## Discussion

In this report, we describe the development of small polyanionic molecules, such as CBS, mCBS, and MTS (mol. wt. ~0.9–1.4 kDa), that are very effective inhibitors of a number of pathological processes mediated by DNA-free histones, such as cytotoxicity, erythrocyte fragility/deformability, and platelet activation in vitro and in vivo. In fact, it seems likely that SPAs mediate their inhibitory activity through rapid precipitation of DNA-free histones in a process similar to protamine neutralization of heparin. Paradoxically, however, we found that MTS, our most potent neutralizer of DNA-free histones in vitro and in vivo, was not effective in vivo against NET-mediated pathologies involving the vascular system, notably cardiac and skin flap IRI and, to a lesser extent, peritonitis-induced sepsis. Subsequently, we found that MTS was also a poor inhibitor of the NET-dependent anti-proliferative activity of sera from sepsis and acute myocardial infarction patients. These data implied that MTS was unable to inhibit NET-mediated pathologies, whereas CBS and mCBS inhibited pathologies induced by both DNA-free and NET-associated histones.

The question remained, what is the molecular basis of structurally similar SPAs being dramatically different in their ability to neutralize NETs. After employing many experimental strategies, we discovered that MTS and heparin were capable of rapidly displacing histones from both native chromatin and NETs causing histone decompaction. This phenomenon has been previously reported for heparin[9,60] and also reflects the well-documented process that occurs during spermiogenesis, wherein histones are globally displaced from sperm DNA by transition proteins, TP1 and TP2, and replaced with the cationic peptide, protamine, that assists in the compaction of chromatin into the sperm head[61]. In contrast to MTS and heparin, mCBS had no such destabilizing effect on chromatin or NETs. Of relevance to our findings is a recent study showing that NETs can occlude blood vessels and cause tissue injury, with two endogenous DNases, DNase1 and DNase1-like 3, being required to solubilize NETs and resolve vascular occlusions[62]. Furthermore, it has been shown that heparin inhibits the activity of DNase113[63]. Also involved in the clearance of NETs are macrophages that endocytose NET structures with partial degradation by DNase1 and opsonization by C1q assisting the process[64]. We have shown that DNase1 degradation of calf-thymus DNA is not inhibited by mCBS, MTS, or heparin (Supplementary Fig. 7). The displacement of histones in NETs by heparin and MTS would be expected to cause unraveling of the DNA structure and, with incomplete degradation by DNases, this may result in the dissemination of NET fragments into the circulation with downstream vascular sequela[65] and reduced capacity for clearance by macrophages. In addition, with the unraveling of the NET structure following histone displacement, exposure of NET-associated antimicrobial

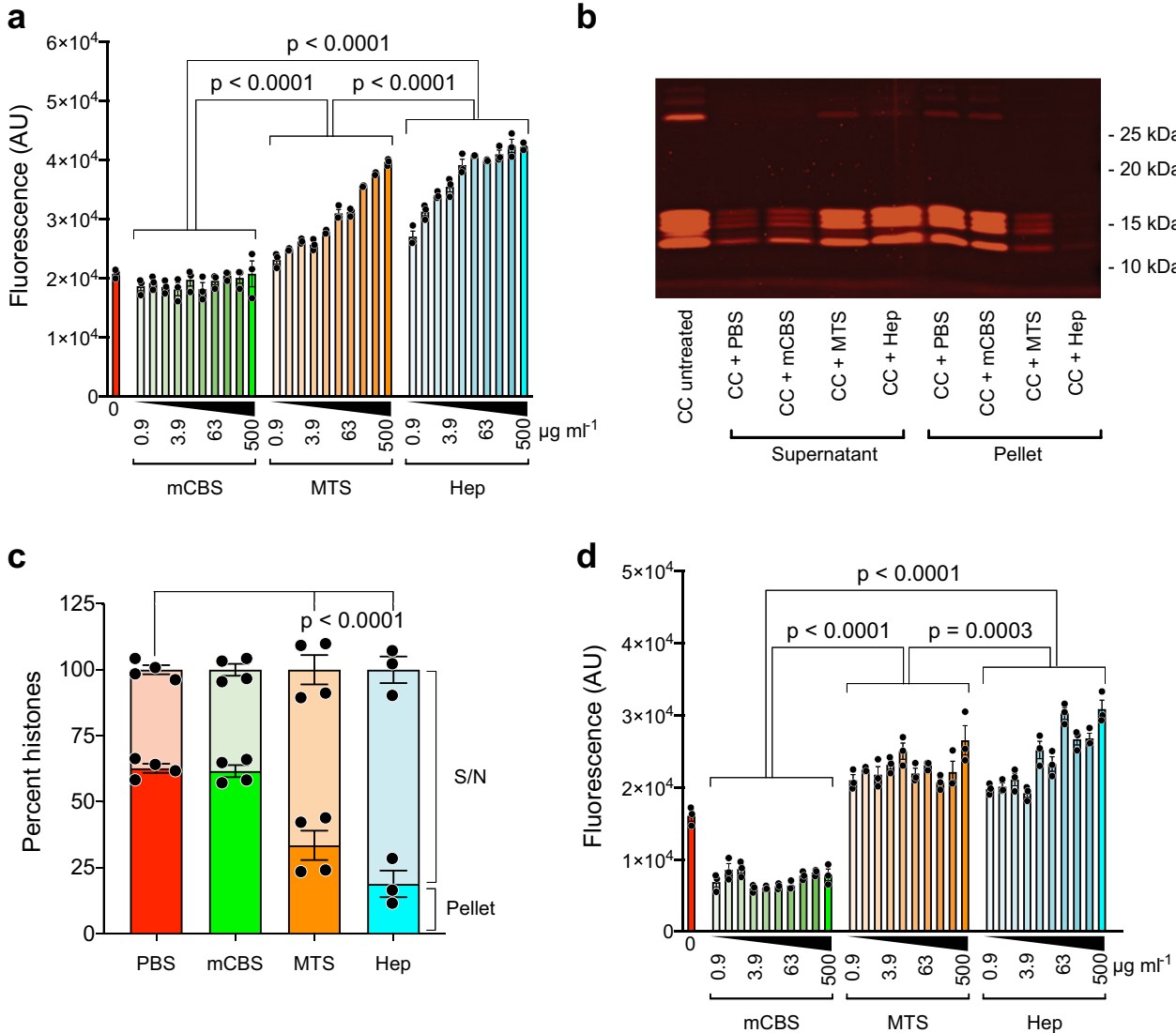

**Fig. 8 Polyanion-induced changes in chromatin and NETs. a** Enhanced uptake of the fluorescent DNA-specific dye, Sytox Green, by chicken chromatin (CC) following exposure to the polyanions MTS and heparin, but not mCBS at concentrations ranging from 0.9 to 500 μg ml$^{-1}$ ($n = 3$/treatment). Data suggest that MTS and heparin can displace histones from CC over a wide concentration range, whereas mCBS cannot. **b** CC exposed to 500 μg ml$^{-1}$ of mCBS, MTS, and heparin, pelleted by centrifugation, and supernatants and pellets collected and run on SDS-PAGE. Representative data from one gel (137% normal size), confirming that MTS and heparin can displace histones from CC, whereas mCBS cannot. **c** Histogram including all the SDS-PAGE data and showing that MTS and heparin displace histones from CC, an effect that is significant ($n = 4$/treatment). **d** Sytox Green uptake by human NETs following incubation with mCBS, MTS, and heparin as in **b** above ($n = 3$/treatment). Data similar to that obtained with CC incubated with MTS and heparin. However, unlike CC, mCBS reduced Sytox Green uptake by NETs by ~50%, suggestive of NET stabilization. Solid circles are individual data points and are presented as mean ± s.e.m. and analyzed by two-way ANOVA with Tukey's correction for multiple comparisons. Source data are provided as a Source Data File.

proteins may result in increased localized vascular damage. The ability of mCBS to neutralize histones but maintain NET integrity is consistent with this model, with the mCBS-bound histones but intact NETs being less damaging in terms of vascular occlusion and vascular injury. A schematic depicting the effect of our SPAs on the pathological effects of extracellular histones is presented in Supplementary Fig. 8.

It should be emphasized that the original aim of this study was to identify a SPA that is as effective as heparin at neutralizing the pathological effects of DNA-free histones, but lacks the undesirable properties of heparin, such as enormous structural diversity, anticoagulant activity, and the ability to bind over 300 different proteins[66]. This aim was achieved with DNA-free histones but the identification of SPAs that inhibit the pathological effects of NET-associated histones proved to be more difficult. In fact, we relied on

in vivo disease models that are thought to be NET dependent to eventually choose mCBS ahead of MTS as the SPA for clinical development. Furthermore, mCBS was chosen ahead of CBS due to the methylation of its reducing terminus, which resulted in mCBS being much more chemically stable than CBS (Supplementary Fig. 3). Also, mCBS has an excellent safety record with, for example, minimal anticoagulant activity, that is, 110-fold lower than LMW heparin and 764-fold lower than unfractionated heparin (Supplementary Figs. 9 and 10 and in "Supplementary information—Properties of mCBS"). In addition, mCBS was well tolerated in rats and dogs when continuously infused at 125 mg kg$^{-1}$ h$^{-1}$ for 14 days (see "Supplementary information—Properties of mCBS"), which is 100–250-fold higher than the maximum dose of heparin recommended for human use. Phase 1 clinical trials in healthy volunteers have also shown that mCBS is very well tolerated in

humans at doses 20-fold higher than the maximum heparin dose administered to patients. In this regard, there have been a number of failed clinical trials in which heparin has been evaluated as a therapy for sepsis[67]. The capacity of heparin to displace histones from NETs, a process that may indirectly cause vascular damage, may provide an explanation for these failures.

Finally, this study provides proof-of-principal data in animals that SPAs are able to inhibit histone-mediated and NET-associated disease states, notably sepsis, cardiac, and skin flap IRI and DVT. It should be noted, however, that in sepsis our SPAs do not inhibit the early, "cytokine storm," stage of the disease but appear to protect against multiple organ failure. For example, we found that our SPAs have little or no effect on LPS-induced sepsis models, which, based on low circulating ALT, LDH, and creatinine levels, do not involve multiple organ failure. In fact, the many failed attempts at developing a treatment for sepsis is probably because they have concentrated on the early "cytokine storm" stage of the disease[68]. Since the main cause of death ≥ patients with sepsis is multiorgan failure, we aim initially to undertake clinical trials with mCBS (Investigational Product Name: STC314) in sepsis. It is hoped that mCBS clinical trials will yield positive results that will underpin the future use of SPAs to treat a wide range of extracellular histone-dependent syndromes, including potentially COVID-19.

## Methods

**Polyanions.** Heparin (porcine mucosa) and the other GAGs listed in Fig. 1 were purchased from Sigma-Aldrich, the exceptions being 15 kDa highly sulfated HS (hi) and under-sulfated HS (lo) that were obtained from Celsus Laboratories. Similarly, the heparin derivatives listed in Fig. 1 were purchased from AMS Bio-technology, with the exception of decarboxylated and *N*- and *O*-desulfated heparin[69,70]. The three cyclic saccharides listed in Fig. 1, β-cyclodextrin sulfate, carboxylated β-cyclodextrin, and β-cyclodextrin phosphate were supplied by Sigma-Aldrich. Sucrose octasulfate was purchased from Cayman Chemicals and we synthesized per-*O*-sulfated cellobiose, maltose, isomaltotriose, panose, raffinose, maltotetraose, maltopentaose, and maltohexaose[46,71]. The remaining mono-, di-, and oligo-saccharides listed in Fig. 1 were purchased in their unsulfated form from Sigma-Aldrich and Carbosyn and fully sulfated as described in the "Supplementary information—Chemistry." The synthesis of sulfated saccharides is not trivial, with uniform sulfation of saccharides being very difficult and a considerable number of reported polysulfated compounds showing substantial heterogeneity, as identified by nuclear magnetic resonance (NMR) spectroscopy and other methods. In fact, co-author von Itzstein has used their recently patented procedure (PCT/AU2018/051338, Priority date: 15 December 2017) developed in his laboratory to synthesize most of the sulfated saccharides used in this study. Their methodology provides uniform sulfation and excellent product homogeneity as demonstrated by NMR spectroscopy (see "Supplementary information—Chemistry"). In some cases, the saccharides were prepared as their corresponding methyl glycoside, prior to sulfation, using a modified generalized glycosidation method (see also "Supplementary information—Chemistry"). The synthesis of mCBS preparations that were *O*-sulfated at 2, 3, 4, and 5 sites was undertaken in the von Itzstein laboratory as described in the "Supplementary information—Chemistry." Phosphomanno-pentaose sulfate (PI-88, aka Muparfostat) was synthesized by Progen Pharma-ceuticals (now TBG Biotechnology) and donated to the laboratory of C.R.P.

In this study, we compared the relative activity of molecules on a gravimetric rather than a molar basis. This decision is justified when dealing with molecules composed of a repeating structure, such as seen with polysaccharides like heparin, which results in each molecule having multiple binding sites for a particular ligand. In cases such as this, molar comparisons give a distorted view of the interaction, giving the larger molecules an exaggerated potency.

**Human subjects.** We have complied with all relevant ethical regulations, and all human-related research being approved by the Australian Capital Territory Health Human Research Ethics Committee. Healthy adult donors were used as a source of neutrophils, erythrocytes, and platelets for in vitro studies. Patients admitted to The Canberra Hospital Intensive Care Unit (ICU) with an APACHE II severity score[72] ≥ 12 on arrival in the ICU and a diagnosis of sepsis were included in our study. Informed consent was obtained from all participants.

Sepsis was diagnosed based on the following criteria[73,74]:

(i)   At least two of the following:

- Tachypnea > 24 b.p.m., or blood gas $PCO_2$ < 32 mm Hg.
- White blood cell count either < 4000 or > 12,000 cells mm$^{-3}$.

- Heart rate > 100 b.p.m.
- Temperature (fever) > 38.0 °C or (hypothermia) < 36.0 °C.

(i)    No alternative cause for systemic inflammatory response syndrome was identified.
(ii)   Evidence of sepsis including positive blood culture, signs of pneumonia on chest x-ray or other imaging.
(iii)  Evidence of end-organ dysfunction: renal failure, liver dysfunction, changes in Glasgow Coma Score (not attributable to other causes), or raised serum lactate.
(iv)   Refractory hypotension requiring inotropic support.

Blood was collected from the cubital fossa on arrival at the ICU, allowed to clot, and the serum collected and stored at −70 °C.

In the case of the acute myocardial infarction study eligibility was based on the following inclusion criteria:

(i)    Must meet at least one of the two following criteria:

- Chest pain (characteristic for acute coronary syndrome), tachycardia (> 99).
- Electrocardiogram changes (ST elevation in two contiguous leads, ST depression in two contiguous leads).
- Raised troponin I (> 0.15 ng$^{-1}$).

(i)    Must not have chronic renal failure/end-stage renal failure/dialysis.
(ii)   Must have evidence of occlusion on angiogram (excludes Takotsubo cardiomyopathy, excludes unstable angina).

Blood (4 ml) was collected prior to percutaneous transluminal coronary angioplasty via the coronary catheter and then 5–10 min post reperfusion of the obstructed coronary vasculature and distal to the ballooned region or stent. Control samples were collected from the cubital fossa of healthy volunteers. Blood samples were allowed to clot and the serum was subsequently removed and stored at −70 °C until analysis was performed.

**Animals.** All animal experiments were approved by the Australian National University Animal Experimentation Ethics Committee and have complied with all relevant ethical regulations regarding the use of research animals. Pathogen-free male and female C57BL/6 mice (6–8 weeks of age), female BALB/c mice (5–6 weeks of age), and male Wistar rats (8–12 weeks old, weighing between 250 and 350 g) were obtained from the Australian Phenomics Facility at the Australian National University. The animals were housed under the following conditions: temperature 18–24 °C, lighting lux 350 lux max, lighting cycle 12:12, humidity 40–70%, and 10–20 air changes per hour.

**Cell lines and cell culture conditions.** HMEC-1, carrying the type O blood group and thus not reactive with anti-blood group antibodies in human sera, were supplied by ATCC and were cultured in the MCDB 131 medium (Gibco) supplemented with 10% heat-inactivated FCS (Sigma), 2 mM L-glutamine (Gibco), 100 IU ml$^{-1}$ penicillin (Gibco), 100 μg ml$^{-1}$ streptomycin (Gibco), 10 ng ml$^{-1}$ epidermal growth factor (Gibco), and 1 μg ml$^{-1}$ hydrocortisone (Sigma-Aldrich). HUVECs were established from primary cultures as previously described[75] and cultured in Medium 199 (Gibco) supplemented with 20% FCS, 2 mM L-glutamine, 100 IU ml$^{-1}$ penicillin, 100 μg ml$^{-1}$ streptomycin, 130 μg ml$^{-1}$ heparin (Sigma-Aldrich), and 1.2 mg ml$^{-1}$ endothelial cell growth supplement (Sigma-Aldrich). CHO-K1 and xylotransferase-1-deficient CHO-K1 cells (pgsA-745 cells), which are HS and GAG deficient, were supplied by ATCC and grown in RPMI-1640 medium supplemented with 5% FCS and antibiotics. All cell lines were incubated in 5% $CO_2$ and ambient $O_2$ at 37 °C and were repeatedly tested for mycoplasma using a MycoAlert Assay Kit (Lonza).

**Histone-mediated cytotoxicity assays.** To determine the cytotoxicity of calf-thymus histones (Sigma-Aldrich), various concentrations of histones (100–800 μg ml$^{-1}$) were added to suspensions of HMEC-1 or HUVEC ($1 \times 10^6$ ml$^{-1}$) in MCDB 131 or Medium 199 medium, respectively, supplemented with 10% heat-inactivated FCS in 96-well flat-bottom plates (NUNC) and incubated for 1 h at 37 °C. Cells were also incubated for the last 15 min at 37 °C with PI (2.5 μg ml$^{-1}$) (Thermo Fisher Scientific), to detect dead cells, and calcein-AM (0.04 μM) (Thermo Fisher Scientific), to detect viable cells, placed on ice, and the percentage of dead and viable cells was determined by flow cytometry using the gating strategies depicted in Supplementary Fig. 11a. In inhibition assays, HMEC-1 were incubated with histones (400 μg ml$^{-1}$) for 1 h at 37 °C in the presence of different concentrations of compounds (12.5–400 μg ml$^{-1}$) prior to the addition of PI and calcein-AM. HMEC-1 cytotoxicity at each compound concentration was then determined based on the formula:

$$\text{Cytotoxicity}(\%) = \frac{\text{Dead (compound and histones)} - \text{Dead (cells alone)}}{\text{Dead (histones alone)} - \text{Dead (cells alone)}} \times 100,$$

and then the half-maximal inhibitory concentration (IC$_{50}$) value for each polyanion was determined based on the line of best fit (nonlinear regression

analysis) using the Prism Software (GraphPad Software). In all assays, MTS ($IC_{50}$ 30 µg ml$^{-1}$) was included as a standard to compensate for experimental variation, with data being adjusted in each assay to MTS having an $IC_{50}$ of 30 µg ml$^{-1}$. In some experiments, suspensions of HMEC-1 were incubated in MCDB 131 medium containing 40% FCS and histones at 800 µg ml$^{-1}$ to assess the effect of high serum levels on the ability of mCBS and MTS to inhibit histone cytotoxicity. Suspensions of HMEC-1 were also depleted of cell surface HS by digestion with either *Flavobacterium* heparinases (HPNSE) I, II, and III (Sigma-Aldrich) or HPSE[76] as reported elsewhere[77], and then examined for sensitivity to histone-mediated cytotoxicity as described above for HMEC-1. Similarly, suspensions of wild-type CHO-K1- and HS/GAG-deficient pgsA-745 CHO-K1 cells were compared for their sensitivity to histone-mediated cytotoxicity.

**Lipid bilayer assays.** Artificial lipid bilayers separated symmetrical 150 or 250 mM KCl (pH ~5.5) solutions[55]. Histones (1 µM, 15.2 µg ml$^{-1}$) were added to bilayers alone or after 0.5–3 h incubation with 10 µM CBS (3.5 µg ml$^{-1}$) or 10 µM MTS (5.1 µg ml$^{-1}$) at ~20 °C. The current was recorded continuously after histone addition until the bilayers broke or the experiment was terminated.

**Calcium flux studies in endothelial cells.** HMEC-1 ($2 \times 10^7$ ml$^{-1}$) in RPMI-1640 medium were incubated with Indo-1 AM (5 µM) (Thermo Fisher)[78,79] at 37 °C for 60 min. After three washes with RPMI-1640 medium supplemented with 5% FCS, the cells were resuspended at $4 \times 10^6$ ml$^{-1}$ in ice-chilled HEPES-buffered saline (NaCl 8 g l$^{-1}$, KCl 0.4 g l$^{-1}$, CaCl$_2$ 0.2 g l$^{-1}$, MgCl$_2$·6H$_2$O 0.2 g l$^{-1}$, D-glucose 1.8 g l$^{-1}$, pH 7.4) supplemented with 10 mM HEPES (Gibco). The cell suspension was kept on ice and used within 3 h. Intracellular Ca$^{2+}$ flux was monitored using flow cytometry (Supplementary Fig. 11d). The cells were pre-equilibrated and maintained at 37 °C during analysis using an external sheath connected to a heated water bath. After the exclusion of cellular debris and clumped cells (on the basis of FSC/SSC light scattering), the basal Ca$^{2+}$ level was monitored for 2 min before the addition of histones in the presence/absence of novel compounds. Ca$^{2+}$ levels were measured at 1, 4, and 10 min post-histone addition with a constant flow rate (~300 events s$^{-1}$). Ca$^{2+}$ flux was determined as an increase in the ratio of geometric mean fluorescence intensity of Ca$^{2+}$-bound over Ca$^{2+}$-unbound Indo-1.

**In vitro erythrocyte microscopy, aggregation, fragility, and deformability assays.** Histone-mediated aggregation of human erythrocytes and its inhibition by various compounds was detected by flow cytometry, based on either FSC and SSC parameters or erythrocyte auto-fluorescence (Supplementary Fig. 11b)[12] and visualized using scanning electron miscroscopy[80]. Similarly, erythrocyte fragility induced by histones, in the presence or absence of inhibitors, was quantified using a sheer stress assay developed in our laboratory[12]. Finally, the reduced deformability of erythrocytes in the presence of histones and the effect of inhibitors on this process was assessed by measuring the passage of erythrocytes through an artificial human spleen consisting of metal microbeads of two sizes in a pipette tip through which treated and untreated erythrocytes were passed using a continuous flow pump (KD Scientific)[50]. Pre- and post-spleen erythrocyte numbers were determined using flow cytometry and the difference was expressed as RBC passage, percentage of control (pre-spleen) value.

**In vitro platelet aggregation and degranulation assays.** For aggregation studies, platelets were isolated from human whole blood collected into Na-citrate vacutainers (Becton-Dickinson) through two-step centrifugation at room temperature ($200 \times g$ for 20 min, then the platelet-rich plasma $800 \times g$ for 15 min); the platelet pellet was resuspended in Hank's balanced salt solution (Gibco) containing calcium and histones added and incubated in the presence/absence of compounds at the concentrations of each as indicated. Samples were assessed for the degree of platelet aggregation after 15 min exposure to histones by flow cytometry using the characteristic log FSC versus log SSC identification of platelets (Supplementary Fig. 11c), with increases in the geometric mean of log FSC indicative of platelet aggregation.

For the platelet activation assay, whole blood collected in Na-citrate vacutainers was monitored for platelet degranulation using the luminescence mode on the Chrono-Log Model 700 with Chrono-Lume reagent (Chrono-Log Corp). Saline (Baxter) (300 µl) was added to pre-warmed blood (420 µl) with a stirrer bar in situ. Chromo-Lume reagent (100 µl) was then added and incubated for 2 min before histones ± compounds diluted in water were added in a total volume of 180 µl at the concentrations indicated. Results were expressed as ATP release and were calculated as a percentage of the histone + saline control.

**In vivo histone toxicity assays.** BALB/c female mice (5–6 weeks of age), which are more prone to histone-induced anemia and easier to inject i.v. at this young age than C57BL/6 mice, were injected i.p. with SPAs at concentrations indicated 10 min prior to i.v. injection of histones (50 mg kg$^{-1}$)[38] in PBS (Sigma-Aldrich). Retro-orbital bleeds were performed with glass Pasteur pipettes 10 min after histone injection and collected blood added to acid citrate dextrose (ACD), the 10 min blood sample being subjected to hematologic analyses for platelet and erythrocyte content using an ADVIA 2120i Hematology Analyzer. Spleens were also harvested at 10 min post-histone injection and splenic Hb content quantified using a Hb

Assay Kit (Sigma-Aldrich). In the case of 4 h blood samples, male C57/BL/6 mice (6–8 weeks of age) were injected with SPAs and histones as above and plasma isolated and stored frozen for subsequent biochemical testing, with markers for liver (ALT), kidney (creatinine), and general tissue (LDH) damage being determined by the Department of Pathology, The Canberra Hospital. To determine the degree of SPAs to inhibit ongoing tissue injury induced by histones, SPAs were injected i.p. 2 h after histones, and 30 min later, plasma was collected and analyzed for ALT, LDH, creatinine, and Hb content. Dead cells were also detected in the liver, lung, and kidneys from these mice by administering i.p. 0.1 ml of a PI solution (0.1 mg ml$^{-1}$) in PBS 5 min prior to blood and organ collection. To obtain reproducible results, a PI stock solution of 1 mg ml$^{-1}$ in PBS was stored frozen in 0.1 ml aliquots and thawed and diluted in 0.9 ml of PBS just prior to use, with leftover PI solutions discarded. Organ slices from PI-injected mice were examined using a Leica DMi6000 automated inverted confocal microscope and the number of fluorescent PI+ cells per field of view determined manually, with at least 4 separate fields/organ being assessed.

**Detection of free histones in plasma.** C57BL/6 mice (6–8 weeks of age) were injected i.v. with histones (50 mg kg$^{-1}$) in normal saline. Retro-orbital bleeds were performed with non-heparinized tubes at 1–2 min post injection and collected in ACD. Isolated plasma from each mouse was then diluted 1/10 in PBS, 200 µl of the diluted plasma added to 100 µl of packed heparin-coupled Sepharose beads (Bio-Strategy) prewashed in PBS in a 0.5 ml Eppendorf tube and the mixture incubated for 30 min at 4 °C on a rotator. The beads were then pelleted by brief centrifugation (~10 s), the supernatant plasma collected, the beads washed twice with PBS, and proteins bound to the beads eluted by boiling in 100 µl of reducing SDS-PAGE sample buffer for 5 min. Eluted proteins were then run on 4–20% tris-glycine gels (Novex), and stained with SYPRO Ruby protein gel stain (Life Technologies). SYPRO Ruby images were captured using the Bio-Rad ChemiDoc Imaging System and Image lab software. ImageJ analysis was used to determine densitometry of histone bands and histone concentrations were determined with reference to histone standards. A 25 kDa unidentified, heparin-binding, protein present in mouse plasma was used as a loading control. When determining the effect of mCBS and MTS on circulating histone levels, experiments were performed as above, except 10 min prior to histone injection mCBS and MTS were injected i.p. at 100 mg kg$^{-1}$.

**Precipitation of histones by polyanions.** Calf-thymus histones (400 µg ml$^{-1}$) were combined with an equal volume of the polyanions mCBS, MTS, or heparin at 400 µg ml$^{-1}$ in PBS, precipitates allowed to form for 5 min at room temperature and then pelleted by centrifugation at $16,100 \times g$ for 10 min. Supernatants were subsequently harvested and polyanion and histone content measured by the 1,9-dimethylmethylene blue (DMMB) (Sigma-Aldrich) assay and the QUBIT® protein assay, respectively. DMMB is a thiazine chromotrope agent that changes its absorption spectrum when bound to sulfated GAGs enabling rapid detection of GAGs in solution[81]. The DMMB assay also readily detects SPAs, such as mCBS and MTS. Supernatants of histone/polyanion precipitates (50 µl) were briefly incubated with 250 µl of DMMB at 16 mg ml$^{-1}$ in glycine-HCl buffer, pH 3.0, in a 96-well flat-bottom microplate. Absorbance was measured at 590 nm in a multimode plate reader (TECAN Infinite M200 PRO). The protein content of supernatants was quantified using the QUBIT® Protein Assay kit (Molecular Probes) according to the manufacturer's instructions. Based on the concentration of polyanions and histones in the supernatants, the percent precipitation of polyanions and histones was calculated and the moles of polyanion precipitated per mole of each histone isoform determined.

**Preparation of partially digested chicken chromatin and soluble human NETs.** Partially micrococcal nuclease-digested chicken RBC chromatin was prepared using the same procedure for preparing nucleosomes from mammalian cell lines[82] with each batch of chicken chromatin being prepared from $10^9$ chicken RBC (Applied Biological Products). In order to obtain partially digested chicken chromatin, isolated chicken chromatin was digested with 2000 gels units ml$^{-1}$ of micrococcal nuclease (New England Biolabs) for 20 min at 37 °C, the reaction being stopped by EDTA addition and the chromatin stored frozen at −20 °C.

For the production of soluble human NETs, initially freshly drawn whole blood from healthy donors was collected in K$_2$EDTA vacutainers® (Becton-Dickinson) and polymorphonuclear neutrophils isolated using the MACSxpress® Human Whole Blood Neutrophil Isolation Kit (Miltenyi Biotec) according to the manufacturer's instructions. Neutrophils were diluted 1:1 in PBS, centrifuged at $300 \times g$ for 10 min and pellets resuspended in medium MCDB 131 (Gibco) supplemented with 0.5% FCS (Sigma). Soluble NETs were prepared as previously described[83] with minor modifications: freshly isolated neutrophils were seeded at $1.5 \times 10^6$ cells well$^{-1}$ in 12-well culture plates (Costar) and stimulated with 50 nM phorbol 12-myristate 13-acetate (Sigma) for 4 h at 37 °C in 5% CO$_2$. Each well was carefully washed twice with 1 ml PBS and then treated with restriction enzyme *Alu*I at 4 U ml$^{-1}$ in 0.4 ml well$^{-1}$ MCDB 131 medium supplemented with 0.5% FCS for 20 min at 37° C in 5% CO$_2$. The supernatant from each well was then collected and centrifuged for 10 min at $300 \times g$ at 4 °C to remove whole cells and debris. The NET-rich supernatants (soluble NETs) were then characterized for their DNA content by briefly incubating 5 µl of soluble NETs with 95 µl of 50 nM Sytox Green

(Invitrogen™) in PBS in a 96-well flat-bottom microplate and measuring green fluorescence at an excitation wavelength of 485 nm and an emission wavelength of 525 nm in a multimode plate reader (TECAN Infinite M200 PRO). Soluble NETs were stored frozen at −20 °C.

**Polyanion-induced changes in chromatin and NETs.** The effect of the polyanions mCBS, MTS, and heparin on the uptake of the fluorescent DNA dye, Sytox Green, by chicken RBC chromatin and soluble NETs was assessed as follows. Each polyanion, starting at 500 µg ml$^{-1}$ and titrating out to 0.9 µg ml$^{-1}$ in PBS, was combined with an equal volume (50 µl) of either chicken chromatin or soluble NETs and immediately 5 µl of each mixture then combined with 95 µl of 50 nM Sytox Green™ (Invitrogen™) in PBS in a 96-well flat-bottom microplate plate and Sytox Green fluorescence measured as above.

Detection of histones released from chicken RBC chromatin following exposure to polyanions was as follows. Chicken RBC chromatin was mixed 1:1 with either PBS, mCBS, MTS, or heparin at 500 µg ml$^{-1}$ final concentration of polyanion, for 10 min at room temperature. The mixtures were subsequently spun in a microcentrifuge at 16,100 × g for 10 min. Supernatants and pellets were collected, and pellets were resuspended to their original volume with PBS. All samples were run on 4–20% tris-glycine SDS-PAGE gels (Novex), and stained with SYPRO Ruby protein gel stain (Life Technologies). SYPRO Ruby images were captured using the Bio-Rad ChemiDoc Imaging System and Image lab software. ImageJ analysis was used to determine densitometry of histone bands.

**DNase inhibition assay.** Calf-thymus DNA (500 ng ml$^{-1}$) (Sigma) was incubated in PBS supplemented with 2 mM CaCl$_2$ and 2 mM MgCl$_2$ alone or in the presence of DNase1 (Roche) (1 µg ml$^{-1}$) and the polyanions heparin, mCBS, and MTS (100 µg ml$^{-1}$) for 10 min at 37 °C in a black 96-well flat-bottom microplate (Nunc). After incubation, the DNA content of samples was quantified by adding 2 µg ml$^{-1}$ of Quant-iT™ PicoGreen® reagent (Molecular Probes) and incubating samples in the dark for 2–5 min. Fluorescence was measured at an excitation wavelength of 485 nm and an emission wavelength of 520 nm in a multimode plate reader (TECAN Infinite M200 PRO).

**Murine DVT model.** Murine DVT model is a modified method[24] in which 8-week old male C57BL/6 mice were anesthetized, a laparotomy incision made, the intestines exteriorized, and then after gentle separation from the abdominal aorta, the inferior vena cava (IVC) immediately below the renal veins was ligated to ~10% patency and all associated IVC tributaries were ligated. The peritoneum and skin were closed following which all mice received an i.v. injection of histones via the tail vein (10 mg kg$^{-1}$) or an equivalent volume of saline followed 5 min later by an i.v. injection of test compounds (50 mg kg$^{-1}$) or saline. Mice were monitored for 48 h, after which they were re-anesthetized, re-opened, and any thrombi that had developed distal to the IVC stenosis were removed for analysis. Sham-operated control animals received laparotomy and 90% ligation of the IVC; however, the ligation was removed immediately after occlusion of the IVC.

**Rat cecal ligation and puncture (CLP) assay for sepsis.** The CLP assay was a modified method performed in male Wistar rats[84]. Test compounds (50 mg kg$^{-1}$) dissolved in saline or an equivalent volume of saline only (Control cohort) were administered i.p. 5 min pre-CLP and 5, 10, and 15 h post op until the cessation of the experiment at 20 h. Sham-CLP rats underwent the same procedure; however, the cecum was not ligated or punctured and these rats received saline at the same times as above. At the conclusion of the experimental time period (20 h) or when morbidity required ethical euthanasia, the rats were anesthetized and blood was collected via cardiac puncture into EDTA for subsequent analysis of liver (ALT) and kidney (creatinine) function by the Department of Pathology, The Canberra Hospital. The propensity for clots to form within the blood samples of the saline-treated control CLP animals (despite the presence of EDTA) prevented successful analysis of plasma samples from all animals.

**Rat cardiac ischemia–reperfusion injury model.** The method used is based on a combination of previously published procedures[85,86]. Male Wistar rats were anesthetized with isofluorane, intubated via tracheostomy, and ventilated with a tidal volume of 1 ml 150 g$^{-1}$ and a respiratory rate of 65 breaths min$^{-1}$. Supplementary oxygen was delivered at a FiO$_2$ of ~30%. A left hemi-thoracotomy was performed to enable visualization of the left ventricle. The left coronary arterial plexus was occluded using an atraumatic snare for 30 min prior to reperfusion for 30 min. Ischemia was confirmed by myocardial hyperemia. The test compounds (30 mg kg$^{-1}$) or an equivalent volume (200 µl) of saline were injected into the lumen of the left ventricle (confirmed with aspiration) 5 min prior to the release of the snare for the reperfusion phase. It should be noted that in this IRI model, rats were subjected to 30 min rather than the more usual 2–3 h reperfusion because we found that reperfusion injury was clearly evident at 30 min, with negligible mortality of reperfused rats, whereas mortality was a significant issue during 2–3 h reperfusion, an effect that potentially could bias results.

At the conclusion of reperfusion (30 min), Thioflavin S (1 ml 200 g$^{-1}$ of body weight) was slowly injected into the lumen of the left ventricle, to define the territory of MVO within the AAR. The AAR territory was determined by the re-occlusion of the atraumatic snare and infusion of blue microspheres into the left ventricle (Unisperse Blue, BASF), and distributed within solutions via ultrasonication using a CD-6800 (Unisonics) sonicator. The heart was then excised from the thorax, rinsed in isotonic saline, and 2 mm sections were cut distal to the atraumatic snare at right angles to the interventricular line. This method produced four myocardial sections that were weighed and photographed (Sony Handycam, Zeiss ×60 optical zoom) under ultraviolet light (territory of MVO) and bright light (AAR territory), before being incubated in tetrazolium chloride to determine the region of necrotic myocardium. Planimetry (ImageJ, Freeware) was used to quantify the areas of the AAR, MVO, and necrosis.

**Rat ischemia–reperfusion tissue-flap model.** Modified method[87] in which male Wistar rats were anesthetized, locally depilated, and a 3 cm × 6 cm fasciocutaneous flap was excised leaving the vascular pedicle intact. The inferior epigastric artery was clamped, a fine rubber sheet was placed under the flap preventing oxygen diffusion from the tissues below, and the flap was re-sutured back into place. The clamp was removed 10 h post application permitting returned blood flow to the flap. Test compounds (50 mg kg$^{-1}$) or saline were administered i.p. 5 min prior to clamp application and 5 min following its removal. The rats were monitored for a total experimental period of 72 h during which rats received additional compound or saline i.p. at 24 and 48 h post op. The "Control No Clamp" rats had the tissue-flap excised and rubber placed underneath prior to re-suturing; however, the vessel was not clamped and they received saline at the same time points as the other rats. At the end of the experimental period, the viability of the flaps was determined by the percentage of black necrotic or reddened areas versus pink viable areas. Despite the application of Elizabethan collars and the use of analgesia as a settling agent, a small number of rats had to be prematurely euthanized when they repeatedly auto-cannibalized their flaps.

**Quantification of DNA in patient sera.** Blood from consenting patients with an APACHE II score >12 upon arrival at ICU was collected from venous access lines into a serum-separator tube, the tubes centrifuged at 2300 × g for 10 min, and the serum harvested and stored at −80 °C. To quantify DNA content, serum from septic patients (5 µl) was incubated with 95 µl of 5 µM Sytox Green (Invitrogen) at 37 °C for 10 min in a 96-well flat-bottom culture plate. Fluorescence intensity was then measured in a fluorescence plate reader (TECAN Infinite M200 PRO) with excitation set at 504 nm and emission at 523 nm.

**Detection and analysis of cytotoxicity of septic and acute myocardial infarction patient sera for endothelial cells.** To measure the cytotoxic activity of septic and myocardial infarction patients sera for endothelial cells in vitro, HMEC-1 were seeded into 96-well plates at 2.5 × 10$^4$ well$^{-1}$ in MCDB 131 medium and grown at 37 °C in 5% CO$_2$ for 48 h prior to the addition of 50 µl of patient serum in the presence/absence of DNase I (10 µg ml$^{-1}$) (Roche), anti-human histone 3 and histone 4 rabbit pAbs (10 µg ml$^{-1}$) (Abcam), or test compounds (200 µg ml$^{-1}$) and incubated for 3 h. $^3$H-thymidine (0.5 µCi, MP Biomedical) was then added to each culture well in 20 µl of HMEC-1 medium and incubated for a further 24 h. At the conclusion of the incubation period, plates were subjected to three freeze/thaw cycles (−70 °C for 30 min, then 37 ± 0.5 °C for 30 min) prior to harvesting and measurement of $^3$H-thymidine incorporation. Using a Filtermate 196 harvester (Packard Bioscience), cell cultures were harvested onto glass fiber filters (EasyTab™-C Self Aligning Filters; Packard Bioscience). Filters were dried at 80 °C and placed in Omnifilter plates (PerkinElmer) and then 20 µl of Microscint-O scintillation fluid (PerkinElmer) was added to each well and the plate sealed with TopSeal-A adhesive film (PerkinElmer). $^3$H-thymidine incorporation was measured using a TopCount NXT™ Microplate Scintillation and Luminescence Counter (Packard Bioscience). Results expressed as a percentage of the proliferation of HMEC-1 not exposed to patient sera

**Rotational thromboelastometry (ROTEM) assays.** Blood was collected from healthy donors into sodium citrate vacutainers and analyzed within 4 h of collection. Histones or test compounds (at the concentrations indicated from a 10 mg ml$^{-1}$ stock solution), or the same volume of water, were added to 330 µl of pre-warmed blood in an Eppendorf tube immediately prior to being loaded into the cup on a ROTEM delta machine (Werfen) as per the manufacturer's instructions and with the type of assay as indicated. An internal water control sample was run alongside three test samples for each assay and the change was calculated as a percentage of the internal control.

**Statistical analysis.** Prism software (GraphPad Software) was used to perform statistical tests and to generate graphs, with details of the test used included in figure legends.

**Reporting summary**. Further information on research design is available in the Nature Research Reporting Summary linked to this article.

## Data availability
Authors can confirm that all relevant data are included in the paper and its Supplementary Information file. Source data are provided with this paper.

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

## Acknowledgements

We thank the following core facilities for their support: Australian Phenomics Facility at ANU for the supply and care of animals used in this study; Imaging and Cytometry Facility at ANU, especially Cathy Gillespie for providing scanning electron microscopy expertise and Michael Devoy and Harpreet Vohra assisting with the flow cytometry studies. We would also like to thank Dr. Teresa Neeman, Biological Data Science Institute, ANU Joint Colleges of Science, Health and Medicine, for her assistance with the statistical analyses. This work was supported by funding from Sirtex Medical Ltd and we also wish to thank Sirtex staff Steve Jones, Marcel Tanudji. and John Patava for their assistance in commercializing our research findings. C.R.P. and L.M.K. received funding from an Australian National Health and Medical Research Council Program Grant (APP1052616). We would also like to acknowledge the companies GlycoSyn Technologies, TetraQ, and ITR Laboratories that contributed to the safety and general evaluation of the properties of STC3141 (mCBS).

## Author contributions

C.H.O., L.A.C., F.K., and B.J.C.Q. each designed, performed, and analyzed in vitro and in vivo experiments, with A.B. and A.M.B. providing technical support. C.F., C.-W.C., P.D.M., G.P., P.C., and M.v.I. synthesized chemical compounds. D.A.S.D. performed the Ca$^{2+}$ flux experiments and E.G., C.S., and A.F.D. undertook the lipid bilayer studies. L.F.A. supervised the rat cardiac IRI experiments and L.F.A. and I.M. provided the serum samples from STEMI and sepsis patients, respectively. D.J.H. performed data analysis, L.M.K. provided advice, and R.W.S. conceived the original idea and provided advice throughout the project. C.R.P. designed and analyzed experiments, conceived ideas, wrote the paper with input from all authors, and supervised the study.

## Competing interests

C.R.P., C.H.O., L.A.C., F.K., B.J.C.Q., A.B., A.M.B., C.F., R.W.S., and M.v.I. have filed patent applications covering the use of SPAs as inhibitors of sepsis. All other authors declare no competing interests.
