## [Peer Review File · Nature Communications]

Reviewers' Comments:

Reviewer #1:

Remarks to the Author:

O'Meara et al provide an interesting and timely study the effect of polyanions on histone-driven cytotoxicity. Although I am in principle quite positive toward sthis study, I have some major comments:

1. Cytotoxicity assays and some of the other in vitro assays were performed in presence of 10% plasma; plasma, however, neutralizes some of the histone effects and hence the authors should perform their experiments with physiological plasma concentrations to study effects of histones and their compounds.
2. The authors study histones in general, but different isoforms have a different ability to activity and kill cells; hence, it is important to use different isoforms throughout the study; also, how was the ratio of the different isoforms in the mix they used?
3. How specific are the polyanions used for histones? would not other NET-bound cationic polypeptides also be neutralized? This is very important for the overall message but also for the possibility to judge possible side-effects of such treatment. A recent paper revealed how modified heparins bind to and neutralize several neutrophil-derived cationic peptides (PMID: 31216421) - the authors should perform similar binding/pull-down assays for their compounds.
4. Throughout the manuscript, the authors use suprphysiological concentrations of histones; I suggest lowering these or providing a rationale for such high concentrations.
5. The novelty of figure 3 is not entirely clear - in fact, the authors disregard a recent study that shows how histones induce cytotoxicity (PMID: 31043745); in addition, this study develops peptides that specifically bind to the N-terminus of histone H4, hereby inhibiting its cytotoxicity; this study should be discussed and the authors should reconsider inclusion of current figure 3.
6. The in vivo assays are not ultimately conclusive; to prove an effect on histone-driven pathologies, the authors need to add groups with histone-neutralizing antibodies and interrogate if the effect of the polyanions is lost in such conditions - unless this is shown the in vivo effects may be due to any cationic molecule.
7. This reviewer does not see the immediate link between figure 7 and the mechanistic studies in this manuscript; why was proliferation and not cell death studies - this does not seem like a good readout within the context of this study.

Reviewer #2:

Remarks to the Author:

In this manuscript, O'Meara and colleagues report the ability of small, persulfonated saccharides to bind to extracellular histones and mitigate their pathogenic activity, at least in part, through histones' release from NETS. The authors present a significant amount of in vitro and in vivo data, most of which is quite interesting, and points (either directly or indirectly) to a role for these persulfonated saccharides in binding to and preventing the activity of free histones. Notably, the team examines a diversity of animal models, strengthening the paper. However, there are a number of shortcomings that should be addressed, if possible.

1. Likely the interaction is governed by sulfate interactions with positive amino acids in the heavily cationic histones. How this is interaction specific? Part of the importance of this question is that persulfonated saccharides are known to interact with quite a few components within serum, not

the least of which is complement and/or the contact system. What complicates this is that rodent species are likely not indicative at all of what would happen upon administration to humans, as was observed with persulfonated chondroitin sulfate.

2. Related to 1, above, the authors make a fairly substantial leap in explaining the disparate results between (m)CBS and MST, where MST appears to be significantly more active *in vitro* and *in vivo* when histones are added exogenously yet has substantially less activity in more indirect animal models of disease. The authors claim this is due to differential binding to DNA-free and NET-associated histones – but, from my reading, this potential explanation is based on the animal findings and not mechanistic studies. Without more data, I find other hypotheses, such as activation/inhibition of other pathways, in a more pleiotropic manner, a more likely explanation. Other, more minor points include –

λAt several points in the manuscript, there is a comparison between the persulfonated compounds and heparin. This comparison is completed on a gravimetric basis and the claim is made that the compounds are more active than heparin. On its face, this is an apples-to-oranges comparison. A more correct comparison would be on a molar basis or on a moles of sulfate basis or something equivalent.

λAt the end of the manuscript, the authors state some of the attributes of mCBS (page 24), including the fact that it is “cheap to manufacture” and “extremely stable”. There is no data to back up these claims. Without data, these claims should be removed.

In my view, the authors need to answer points 1 and 2 prior to consideration.

Reviewer #3:

Remarks to the Author:

1. A key shortfall of this work is that the concept of using anionic molecules to neutralize histone toxicity is not new. Mishra et al. described the interaction of polysialic acid with histone H1 (Mishra B et al, *J Neurosci.* 2010 Sep 15;30(37):12400-13). Galuska/Preissners group showed that polysialic acid inhibits histone toxicity (Saffarzadeh M et al. *PLoS One.* 2012;7(2):e32366.) and investigated its therapeutic potential in several recent papers (e.g. Zlatina K et al. *Int J Mol Sci.* 2018 Jun 5;19(6)., Galuska CE et al. *Front Immunol.* 2017 Sep 29;8:1229, Galuska SP et al. *FEBS J.* 2017 Jun;284(11):1688-1699). This body of literature should be discussed in the present manuscript.

2. Another shortfall is the lack of data on off-target effects of SPAs, for example regarding circulating SPA-histone complexes, or binding to other targets. Thrombin generation assays should be performed, or tail bleeding times.

3. The importance of NETs in various diseases is described in the introduction; however, mostly animal model papers are cited, although manuscripts presenting human data are available (e.g. Savchenko AS et al. *J Thromb Haemost.* 2014 Jun;12(6):860-70; Mangold A et al. *Circ Res.* 2015 Mar 27;116(7):1182-92; Pertiwi KR et al. *Thromb Haemost.* 2018 Jun;118(6):1078-1087.)

4. Do SPAs compete with heparins? What is the incremental effect of SPAs over LMWH or DNase? It was shown that efficient degradation of extracellular chromatin by DNases is crucial for homeostasis (Jiménez-Alcázar et al. *Science.* 2017; 358, 1202–1206). It was described that heparin has contrasting effects on DNase1 and DNase1L3 activity (Napirei et al. *FEBS Journal.* 2009; 276; 1059–1073). The effect of SPAs on the capacity of DNases to degrade DNA/chromatin needs to be sorted out.

5. Authors screened for small polyanions (SPAs)(~0.9-1.4 kDa) that interact electrostatically with histones, neutralizing their pathological effects. How exactly is the mechanism of action?

6. Heparin toxicity needs to be explained in more detail. In humans exorbitant heparin doses may cause bleeding but are not toxic.

7. The terms erythrocyte fragility/deformability and aggregation need to be defined. For example, how does RBC aggregation occur? Aggregation is well studied in platelets, but less so in RBC. Are histones binding to RBC surfaces? The authors demonstrate that loss of cell surface GAGs has little

effect on cytotoxicity. Therefore, which other RBC surface epitope are histones binding to? Or do they rupture the RBC lipid bilayer?

8. Authors point out the difference of CBS/mCBS and MTS potency towards DNA-free and NET-associated histones. This difference is derived from in vivo data. In vitro, SPAs were only tested against DNA-free histones. Activity of SPA's against NET-associated histones should also be investigated in vitro.

minor points:

1. broadly-inhibitory, non-toxic SPA.. What makes authors not call the SPA inhibitory for histones. What is broadly?

2. please correct: lowly and highly sulfated

3. Please rephrase: MTS is ineffectual against NET-associated histones

4. Crea needs to be spelled out

5. Please, add dots in Figure 7g

6. The application route (iv, ip, intraventricular) and the dose (10, 30, 50 mg/kg) varies considerably between the used animal models. If caused by limitations of the model, this should be mentioned; otherwise, comparable routes and doses should be used.

7. The authors tested the effect of patient sera on HMEC proliferation in ex vivo assays. Why not in other in vitro experiments? Especially cytotoxicity would be interesting.

8. From Figure 5 on authors do not indicate what the figures exactly plot; presumably, mean \pm s.e.m. is displayed. This might not be appropriate for every experiment, especially in vivo data.

Reviewer #4:

Remarks to the Author:

In this paper O'Meara et al have described the development of small polyanionic (SPAs) molecules as inhibitors of several pathologies that are associated with, and probably exacerbated by DNA-free histones. They went on to identify several SPAs, such as mCBS, CBS and MTS, that inhibit pathologies related to DNA-free and NET-associated histones or selectively inhibit DNA-free histones. This is a novel and interesting paper that provides important information about the possible therapeutic value of these SPAs, which are small poly sulfated di- or trisaccharides. However, several issues that were not well addressed in the paper should be considered by the authors, as follows:

1- The authors state in the Discussion that "Phase 1 clinical trials in healthy volunteers have also shown that mCBS is very well tolerated in humans at doses 20-fold higher than the maximum heparin dose administered to patients." Given that the current paper appears to be meant to describe the discovery of such SPAs as mCBS, it is confusing to hear that it is already in clinical trials. Moreover, this sentence was not referenced by the authors. Accordingly, a better description of this point is important, as it could impact the novelty of this manuscript.

2- Related to the comment above, the authors also state in the Discussion that "...mCBS was well tolerated in rats and dogs when continuously infused at 12.5 mg kg⁻¹ h⁻¹ for 14 days (unpublished data), which is 100 to 250-fold higher than the maximum dose of heparin recommended for human use." This unpublished data needs to be presented in the current paper. In fact, this statement brings up many other things that need to be addressed by the current paper, such as a description of pharmacokinetics and dynamics, i.e. how long-lived are these small SPAs when injected, as in the current study, what is their clearance rate, how does this related temporally to their beneficial effects?

3- It was important for the authors to compare their SPAs to heparin, in terms of efficacy in the pathology models, as well as in terms of untoward effects. One important effect I wondered about, even though acute administration of the SPAs did not increase platelet aggregation, is the pro-thrombin or clotting times after acute, vs chronic administration?

4- In terms of the cardiac IRI experiments shown in Fig. 6c the authors use IZ terminology to mean ischemic zone, but in IRI nomenclature "I" usually refers to infarct zone. A better terminology would be area at risk (AAR). And the authors used the term MVO for microvascular obstruction, and separately show infarct territory. It's fine to show MVO but description of the infarct size should be just that, and is should be calculated as Infarct Area as a % of the AAR. MVO is a separate determination, and interesting to note, but not the same as infarct size. Furthermore, the authors should show visual examples of a CBS or MTS treated animal heart in addition to the saline treated heart they show.

5- Again, related to Fig. 6c, I'm not sure how the CBS vs MTS data shown here, where CBS was more effective than MTS and reducing infarct size, relates to Extended Data Fig. 2, where the right side of the figure shows MTS to be more potent or powerful than CBS in the ischemia models?

6- Also, What are the sera concentration of histones in the damage models and how are they affected by the supplementation of SPAs?

7- The authors should demonstrate in vivo evidence for the SPAs binding to histones if this is the presumed mechanism of protection and the kinetics of this binding, because SPAs were injected 5-minutes before reperfusion in cardiac IRI model and sacrificed only 30-minutes later implying rapid effect.

8- The authors should describe the rationale for such an acute cardiac IRI model (i.e. 30-min occlusion; 30-min reperfusion). Typically, in the literature the most acute timeframe of examining reperfusion injury where NETs was the focus is 3h, a time where the infarct is believed to be mostly due to intracellular ROS generation and calcium dysregulation in cardiac myocytes.

9- The authors should measure plasma cTnI levels upon cardiac IRI with and without SPA administration to delineate between protection from necrosis or protection that is secondary in action, as the authors propose.

10- The authors should assess the effects of SPAs after more chronic cardiac IRI models, such as the 45 days post IRI injury in some of the authors' references, a time when functional benefit can be examined.

Rebuttal of Reviewers' Comments

NCOMMS-19-40206-T-Neutralizing the pathological effects of extracellular histones with small polyanions

Note new/altered text and Figures in revised manuscript are indicated by yellow highlighter pen.

Before responding to the reviewers' comments, we would like to briefly highlight our recent research findings that are highly relevant to their concerns:

- (i) Following further extensive investigations, we now understand why two small polyanions (SPAs), mCBS and MTS, despite both being able to neutralize DNA-free histones, differ markedly in their response to NET-associated histones. The key new finding is that MTS (and heparin) induce the rapid displacement of histones from chromatin and NETs while mCBS has no such effect. This finding and its functional implications are presented in detail in a new "Mode of action of SPAs" section in the revised manuscript, in the second paragraph of the Discussion and in a new figure (Fig. 8).
- (ii) We have found that mCBS and MTS eliminate DNA-free histones by rapidly precipitating them from solution and, *in vivo*, this results in clearance of histones from the circulation within 1-2 minutes. This mechanism is analogous to protamine interacting with heparin and neutralizing its anticoagulant activity (see new "Mode of action of SPAs" section and new Supplementary Figs 4 and 5).
- (iii) The speed and specificity of the interactions outlined in (i) and (ii) above highlights the fact that we are not simply dealing with non-specific binding of SPAs to a range of cationic proteins.

Reviewer #1 (Remarks to the Author):

O'Meara et al provide an interesting and timely study the effect of polyanions on histone-driven cytotoxicity. Although I am in principle quite positive toward this study, I have some major comments:

1. Cytotoxicity assays and some of the other in vitro assays were performed in presence of 10% plasma; plasma, however, neutralizes some of the histone effects and hence the authors should perform their experiments with physiological plasma concentrations to study effects of histones and their compounds.

Culturing cells in 100% serum is not practicable as it provides no room for the addition of cells, media or various molecules with different solubilities. However, we decided that 40% serum was a physiologically relevant concentration to use as the protein content of interstitial fluids, which many cells encounter, has been reported to be ~40% that of plasma (REF 49). When we compared the effect of different concentrations of serum (10% and 40%) on the ability of the small polyanions (SPAs) mCBS and MTS to inhibit histone cytotoxicity for endothelial cells, the results obtained were very similar. The only difference was that the concentration of histones needed to be doubled in the 40% serum cultures in order to obtain the same level of histone cytotoxicity as

that observed in 10% serum. This is not surprising as it is well known that there are a considerable number of histone binding proteins in serum. Based on these findings there appears to be no justification for repeating all the histone cytotoxicity experiments at very high serum concentrations, an approach that would also be very expensive. These new data are presented in a new Figure (Supplementary Fig. 2) and discussed in the text (Supplementary File, page 2).

2. The authors study histones in general, but different isoforms have a different ability to activity and kill cells; hence, it is important to use different isoforms throughout the study; also, how was the ratio of the different isoforms in the mix they used?

We are acutely aware of different histone isoforms having different abilities to kill cells, with histones H3 and H4 usually being more cytotoxic than H2A and H2B. In fact, one of the earliest studies we attempted in this project was to assess the ability of SPAs to inhibit the membrane damaging effects of different histone isoforms. Unfortunately, using commercially available recombinant histone isoforms, we were unable to get reproducible results, with solubility of the histones, particularly H3 and H4, being a major issue. Our research institute (JCSMR) houses some international nucleosome experts who were not at all surprised to hear of our problem as isolated histone isoforms are notoriously insoluble and they questioned the logic of using them citing that histone sub-units readily form the octamer without the need for DNA or other protein involvement and, hence, are not normally found in isolation. H3 and H4 form a tetramer while H2A and H2B form a dimer, these initial interactions being hydrophobic, while subsequent weaker interactions form the octamer (Eikbush & Moudrianakas *Biochemistry*, 1978). Thus, after spending \$10,000 on recombinant histones with little to show for it, we reluctantly ceased research in this area.

The ratio of different histone isoforms in chicken red blood cell chromatin, in calf thymus histone and in human NET preparations was determined by SDS-PAGE and staining gels with the Sypro Ruby protein gel stain. In both the chicken chromatin and calf thymus histone preparations, histones H2A, H2B, H3 and H4 could be easily identified but calf H1 and the chicken H1 homologue, H5, stained weakly compared to the other 4 histone isoforms. In contrast, the NET-associated histones were substantially degraded with only two lower mol. wt. bands being detected and too little sample to detect a H1 band (new Supplementary Fig. 6).

3. How specific are the polyanions used for histones? would not other NET-bound cationic polypeptides also be neutralized? This is very important for the overall message but also for the possibility to judge possible side-effects of such treatment. A recent paper revealed how modified heparins bind to and neutralize several neutrophil-derived cationic peptides (PMID: 31216421) - the authors should perform similar binding/pull-down assays for their compounds.

As already noted in our manuscript (Results, 3rd paragraph) previous studies with small polyanions (SPAs), like mCBS and MTS, have shown that they do not normally bind to cationic enzymes, growth factors, chemokines and coagulation factors. Sulfated pentasaccharides are usually the minimum polyanionic structure required for strong binding of cationic proteins, negligible binding to sulfated di- and tri-saccharides being reported. The exception to this rule, however, is histones that, due to their requirement to package DNA into nucleosomes, are exceptionally cationic and are shown in this

manuscript to be able to strongly bind sulfated disaccharide and sulfated trisaccharide molecules.

Further evidence of the potency of the interaction between histones and SPAs comes from the observation that addition of mCBS and MTS to histones in solution results in extremely rapid precipitation of the histones, this effect occurring even in the presence of serum. In contrast, addition of mCBS and MTS to plasma alone results in no precipitate. Similarly, when mCBS and MTS were injected i.p. into mice 10 min prior to i.v. injection of histones and plasma samples collected 1-2 min later, MTS produced a substantial and highly significant decline in histone levels (~80%) whereas mCBS had a smaller, but not statistically significant, effect (see new Supplementary Fig. 5a, b). Thus, *SPAs rapidly eliminate histones* from plasma and this represents a mechanism of action of these compounds *similar to the anticoagulant activity of heparin being neutralized by protamine*. Experiments were undertaken to quantify this precipitation reaction *in vitro* and are depicted in new Supplementary Fig. 4 and discussed in the 'Mode of Action of SPAs' section in the revised manuscript.

The low toxicity of SPAs is also consistent with their limited reactivity with proteins other than histones. In fact, the pharmacology and toxicology of mCBS, our chosen SPA for clinical development, has been extensively investigated and only weak anticoagulant activity (i.e., 110-fold lower than LMW-heparin and 764-fold lower than heparin) has been identified as a possible dose limiting toxicity (see Supplementary Figs 9 and 10 and 'Anticoagulant Properties of mCBS.Na' section in Supplementary Information – Properties of mCBS.Na). Detailed histological examination of animals receiving very high and prolonged doses of mCBS (i.e., 2785 mg kg⁻¹ day⁻¹ for 14 days) revealed vacuolation, particularly in cells in the liver and kidney, as well as the appearance of foamy macrophages, most likely due to the phagocytosis and clearance of very high levels of mCBS (Supplementary Information – Properties of mCBS.Na). However, these effects were seen at mCBS doses more than 100-fold higher than would ever be administered to patients.

Finally, binding/pull down experiments are difficult to perform with mCBS and MTS as they involve the attachment of bulky reporter molecules (i.e., biotin/fluorescein) that may interfere with the interaction of histones with mCBS/MTS. In fact, in attempts to improve the half life of CBS we have replaced the methyl group on the reducing terminus of CBS with a range of hydrophobic groups, some of these substitutions markedly reducing the ability of CBS to neutralize histone cytotoxicity.

4. Throughout the manuscript, the authors use supraphysiological concentrations of histones; I suggest lowering these or providing a rationale for such high concentrations.

The histone concentrations we have used in our experiments (i.e., up to 400 µg ml⁻¹) are not supra-physiological but at the upper end of the concentration of circulating histones observed in sepsis, pancreatitis and trauma patients (REFS 6, 14, 17). In fact, the major challenge in controlling these syndromes is having therapies that effectively neutralize the high levels of circulating histones without having significant side-effects.

5. The novelty of figure 3 is not entirely clear - in fact, the authors disregard a recent study that shows how histones induce cytotoxicity (PMID: 31043745); in addition, this

study develops peptides that specifically bind to the N-terminus of histone H4, hereby inhibiting its cytotoxicity; this study should be discussed and the authors should reconsider inclusion of current figure 3.

We thank the reviewer for seeking clarification. The text of the revised manuscript has been modified to include discussion of the histone H4 paper (Manuscript File, page 12, lines 1-2). The novelty of our Figure 3 is not that it shows that histones can directly induce pore formation in lipid bilayers (this has been known for some time) but that our **SPAs inhibit histones from inducing pore formation**. The revised text has been changed slightly to make this point clearer.

6. The in vivo assays are not ultimately conclusive; to prove an effect on histone-driven pathologies, the authors need to add groups with histone-neutralizing antibodies and interrogate if the effect of the polyanions is lost in such conditions - unless this is shown the in vivo effects may be due to any cationic molecule.

We agree, in theory, that histone neutralizing antibodies could be used to unequivocally demonstrate that the action of our SPAs is histone-specific, however, such antibodies were not available when this work was undertaken and what the Reviewer requests is a repetition of years of work. The additional extensive in-vitro and in-vivo investigations we have included in this resubmission demonstrate the specificity of the interaction of the SPAs for free- and NET-bound histones and, along with previous publications (REF 46-48), our SPAs would be expected to have minimal off-target effects as they interact weakly, or not at all, with a limited number of cationic proteins, the very weak anticoagulant activity of our lead drug, mCBS, being the case in point (see point 3 above). Also, as mentioned earlier, **histones are rapidly and selectively precipitated by SPAs in a process similar to protamine precipitating and neutralizing heparin**.

7. This reviewer does not see the immediate link between figure 7 and the mechanistic studies in this manuscript; why was proliferation and not cell death studies - this does not seem like a good readout within the context of this study.

We found that when HMEC-1 were exposed to 50% serum from sepsis patients the Calcein-AM and PI fluorescent staining method we used to differentiate live and dead cells throughout this study (Supplementary Fig. 1) gave variable results. The reason for this variability is unclear but was associated with poor staining and cell recovery. Furthermore, LDH release could not be used as a measure of cell death as sera from sepsis patients often have high circulating LDH levels. In contrast, incorporation of ³H-thymidine into the DNA of dividing HMEC-1 gave very reproducible results over relatively prolonged incubation periods, i.e., 24 h. Thus, we had no alternative but to use inhibition of proliferation of HMEC-1, based on ³H-thymidine incorporation, as an indirect measure of the cytotoxicity of sepsis patients' sera for HMEC. Text describing this issue is incorporated into the revised manuscript (Manuscript File, page 18, paragraph 1, lines 1-7).

Reviewer #2 (Remarks to the Author):

In this manuscript, O'Meara and colleagues report the ability of small, persulfonated saccharides to bind to extracellular histones and mitigate their pathogenic activity, at

least in part, through histones' release from NETS. The authors present a significant amount of in vitro and in vivo data, most of which is quite interesting, and points (either directly or indirectly) to a role for these persulfonated saccharides in binding to and preventing the activity of free histones. Notably, the team examines a diversity of animal models, strengthening the paper. However, there are a number of shortcomings that should be addressed, if possible.

1. Likely the interaction is governed by sulfate interactions with positive amino acids in the heavily cationic histones. How this is interaction specific? Part of the importance of this question is that persulfonated saccharides are known to interact with quite a few components within serum, not the least of which is complement and/or the contact system. What complicates this is that rodent species are likely not indicative at all of what would happen upon administration to humans, as was observed with persulfonated chondroitin sulfate.

Refer to response to similar question by Reviewer #1, point 3.

We agree that persulfonated **polysaccharides**, such as persulfonated chondroitin sulfate, are **high mol. wt. (~20-40 kDa) molecules** that interact with quite a few components within serum and can mediate species-specific side effects. Importantly, this is not the case with SPAs, such as **sulfated di- and tri-saccharides, which are very low mol. wt. (0.9-1.4 kDa) molecules** (REF 46-48). Also, our chosen drug for clinical development, mCBS, has already undergone extensive preclinical pharmacology and toxicology and a successful Phase 1 clinical trial in healthy volunteers. For more details see 'Supplementary Information – Properties of mCBS.Na'.

2. Related to 1, above, the authors make a fairly substantial leap in explaining the disparate results between (m)CBS and MST, where MST appears to be significantly more active in vitro and in vivo when histones are added exogenously yet has substantially less activity in more indirect animal models of disease. The authors claim this is due to differential binding to DNA-free and NET-associated histones – but, from my reading, this potential explanation is based on the animal findings and not mechanistic studies. Without more data, I find other hypotheses, such as activation/inhibition of other pathways, in a more pleiotropic manner, a more likely explanation.

Recently we identified a striking difference in the effect of mCBS and MTS on native chromatin and NETs. While MTS (and heparin) induce the rapid displacement of histones from chromatin and NETs, mCBS has no such effect. This novel finding is presented in detail in the new "Mode of Action of SPAs" section in the manuscript, in the second paragraph of the Discussion and in a new Figure (Fig. 8).

Other, more minor points include –

At several points in the manuscript, there is a comparison between the persulfonated compounds and heparin. This comparison is completed on a gravimetric basis and the claim is made that the compounds are more active than heparin. On its face, this is an apples-to-oranges comparison. A more correct comparison would be on a molar basis or on a moles of sulfate basis or something equivalent.

Comparing the relative activity of molecules on a gravimetric rather than a molar basis is justified when dealing with molecules composed of a repeating structure, such as seen with polysaccharides like heparin, which results in each molecule having multiple binding sites for a particular ligand. In cases such as this, molar comparisons give a distorted view of the interaction, giving the larger molecules an exaggerated potency. This point has been included in the Methods section of the manuscript (Manuscript File, page 32, paragraph 1).

At the end of the manuscript, the authors state some of the attributes of mCBS (page 24), including the fact that it is “cheap to manufacture” and “extremely stable”. There is no data to back up these claims. Without data, these claims should be removed.

The cheap to manufacture text has been deleted and the stability of the compounds mCBS versus CBS is shown in a new Supplementary Fig. 3 and in the ‘Supplementary Information – Properties of mCBS.Na’ file.

In my view, the authors need to answer points 1 and 2 prior to consideration.

Reviewer #3 (Remarks to the Author):

1. A key shortfall of this work is that the concept of using anionic molecules to neutralize histone toxicity is not new. Mishra et al. described the interaction of polysialic acid with histone H1 (Mishra B et al, J Neurosci. 2010 Sep 15;30(37):12400-13). Galuska/Preissners group showed that polysialic acid inhibits histone toxicity (Saffarzadeh M et al. PLoS One. 2012;7(2):e32366.) and investigated its therapeutic potential in several recent papers (e.g. Zlatina K et al. Int J Mol Sci. 2018 Jun 5;19(6)., Galuska CE et al. Front Immunol. 2017 Sep 29;8:1229, Galuska SP et al. FEBS J. 2017 Jun;284(11):1688-1699). This body of literature should be discussed in the present manuscript.

We agree that the use of anionic molecules to neutralize histone cytotoxicity is not new, with heparin and polysialic acid being shown previously to inhibit histone cytotoxicity. **In fact, the ability of polyanions to neutralize histone cytotoxicity is the theoretical basis of this study.** Unfortunately, in the introduction to our manuscript we discussed the use of heparin as a neutralizer of histone cytotoxicity but did not mention polysialic acid, an omission that has now been corrected in the Results section of the revised manuscript (Manuscript File, page 4, paragraph 2, lines 1-4). In the case of heparin, however, it is a potent anticoagulant and reacts with many other proteins not involved in coagulation. Thus, unlike polysialic acid, heparin in its native state is not an attractive therapeutic for neutralizing histones *in vivo*, a problem that this study attempts to solve by using SPAs.

2. Another shortfall is the lack of data on off-target effects of SPAs, for example regarding circulating SPA-histone complexes, or binding to other targets. Thrombin generation assays should be performed, or tail bleeding times.

In this study we have attempted to identify small polyanions (SPAs) that neutralize histone cytotoxicity as effectively as heparin but lack many of the off-target effects of

heparin, particularly anticoagulant activity. Having worked with heparan sulfate mimetics as potential polyanion-based therapeutics for over 30 years we knew already that SPAs, such as sulfated di- and tri-saccharides, exhibit negligible inhibitory activity in a wide range of heparan sulfate-dependent processes (see REF 46-48). In contrast, we found some SPAs, namely CBS, mCBS and MTS, strongly interact with histones. Also, the anticoagulant activity of our lead drug, mCBS, has been extensively studied using a wide range of *in vitro* coagulation assays and shown to have very low anticoagulant activity when administered to animals and humans (see Supplementary Figs 9 and 10 and ‘Anticoagulant Properties of mCBS.Na’ section in Supplementary Information – Properties of mCBS.Na).

Further evidence of the potency of the interaction between histones and SPAs comes from the observation that addition of mCBS and MTS to histones in solution results in extremely rapid precipitation of the histones, this effect occurring even in the presence of serum. In contrast, addition of mCBS and MTS to serum alone results in no precipitate. Similarly, when mCBS and MTS were injected i.p. into mice 10 min prior to i.v. injection of histones and plasma samples collected 1-2 min later, MTS produced a substantial and highly significant decline in histone levels (~80%) whereas mCBS had a smaller, but not statistically significant, effect (see new Supplementary Fig. 5a, b). Thus, ***SPAs rapidly eliminate histones*** from plasma and this represents a mechanism of action of these compounds ***similar to the anticoagulant activity of heparin being neutralized by protamine***. Experiments were undertaken to quantify this precipitation reaction *in vitro* and are depicted in Supplementary Fig. 4 and discussed in the ‘Mode of Action of SPAs’ section in the revised manuscript.

3. The importance of NETs in various diseases is described in the introduction; however, mostly animal model papers are cited, although manuscripts presenting human data are available (e.g. Savchenko AS et al. J Thromb Haemost. 2014 Jun;12(6):860-70; Mangold A et al. Circ Res. 2015 Mar 27;116(7):1182-92; Pertiwi KR et al. Thromb Haemost. 2018 Jun;118(6):1078-1087.)

Thank you for pointing this out. In the revised manuscript, human studies demonstrating the importance of NETs in various diseases have been referred to as suggested (Manuscript File, page 3, paragraph 2).

4. Do SPAs compete with heparins? What is the incremental effect of SPAs over LMWH or DNase? It was shown that efficient degradation of extracellular chromatin by DNases is crucial for homeostasis (Jiménez-Alcázar et al. Science. 2017; 358, 1202–1206). It was described that heparin has contrasting effects on DNase1 and DNase1L3 activity (Napirei et al. FEBS Journal. 2009; 276; 1059–1073). The effect of SPAs on the capacity of DNases to degrade DNA/chromatin needs to be sorted out.

A major advantage of SPAs is that they can be safely administered at very high levels, much higher than can be safely achieved with heparin and LMWH. For example, mCBS has minimal anticoagulant activity, it being 110-fold lower than LMW-heparin and 764-fold lower than unfractionated heparin (Supplementary Fig. 9, 10 and in ‘Supplementary Information – Properties of mCBS.Na’). In addition, mCBS was well tolerated in rats and dogs when continuously infused at 125 mg kg⁻¹ h⁻¹ for 14 days (see ‘Supplementary Information – Properties of mCBS.Na’), which is 100 to 250-fold higher than the maximum dose of heparin recommended for human use. Thus, these

differences between SPAs and LMWH/heparin are much more than incremental.

Additional experiments were undertaken evaluating the ability of DNaseI to degrade chromatin in the presence of the SPAs and heparin. These data are presented in the Supplementary section. Additionally, the role of DNase1 and DNase113 in the biological effects of mCBS, MTS and heparin is considered in the Discussion, 2nd paragraph.

5. Authors screened for small polyanions (SPAs)(~0.9-1.4 kDa) that interact electrostatically with histones, neutralizing their pathological effects. How exactly is the mechanism of action?

The mechanism of action of SPAs interacting with DNA-free histones is described in a new section of the manuscript entitled 'Mode of action of SPAs' (Manuscript File, pages 21-23).

6. Heparin toxicity needs to be explained in more detail. In humans exorbitant heparin doses may cause bleeding but are not toxic.

The description of heparin toxicity has been changed to indicate that, indeed, the mice needed to be terminated due to bleeding (Manuscript File, page 15, paragraph 1, line 9).

7. The terms erythrocyte fragility/deformability and aggregation need to be defined. For example, how does RBC aggregation occur? Aggregation is well studied in platelets, but less so in RBC. Are histones binding to RBC surfaces? The authors demonstrate that loss of cell surface GAGs has little effect on cytotoxicity. Therefore, which other RBC surface epitope are histones binding to? Or do they rupture the RBC lipid bilayer?

We have found that fluorescein-labelled histones bind to the surface of RBC, as they do to all cell types tested, and act as cross-linkers of RBC. Furthermore, our finding that histones can destabilize artificial lipid bilayers (Fig. 3c), a process our SPAs can inhibit, is consistent with histones directly interacting with RBC membranes and increasing their fragility.

8. Authors point out the difference of CBS/mCBS and MTS potency towards DNA-free and NET-associated histones. This difference is derived from in vivo data. In vitro, SPAs were only tested against DNA-free histones. Activity of SPA's against NET-associated histones should also be investigated in vitro.

We have generated new *in vitro* data that reveals how mCBS and MTS differ dramatically in their response to chromatin and NET-associated histones. These data are presented in detail in a new section in the revised manuscript entitled 'Mode of action of SPAs' (Manuscript File, pages 21-23).

minor points:

1. broadly-inhibitory, non-toxic SPA.. What makes authors not call the SPA inhibitory for histones. What is broadly?

“Broadly-inhibitory” was referring to an SPA that inhibits both free and NET-

associated histones. To avoid confusion the text, which was in the abstract, has been changed.

2. *please correct: lowly and highly sulfated*

Corrected to 'under' sulfated

3. *Please rephrase: MTS is ineffectual against NET-associated histones*

The sentence has been rephrased.

4. *Crea needs to be spelled out*

This has been corrected to creatinine.

5. *Please, add dots in Figure 7g*

This panel has been removed as, upon reflection, it was not a significant result and did not add any value to this figure.

6. *The application route (iv, ip, intraventricular) and the dose (10, 30, 50 mg/kg) varies considerably between the used animal models. If caused by limitations of the model, this should be mentioned; otherwise, comparable routes and doses should be used.*

The doses and injection routes used were based on previously published models.

7. *The authors tested the effect of patient sera on HMEC proliferation in ex vivo assays. Why not in other in vitro experiments? Especially cytotoxicity would be interesting.*

The reasons for measuring HMEC-1 proliferation rather than cytotoxicity are explained under Reviewer #1, comment 7.

8. *From Figure 5 on authors do not indicate what the figures exactly plot; presumably, mean \pm s.e.m. is displayed. This might not be appropriate for every experiment, especially in vivo data.*

The omission of mean \pm s.e.m. has been corrected in Figure 5 onwards.

Reviewer #4 (Remarks to the Author):

In this paper O'Meara et al have described the development of small polyanionic (SPAs) molecules as inhibitors of several pathologies that are associated with, and probably exacerbated by DNA-free histones. They went on to identify several SPAs, such as mCBS, CBS and MTS, that inhibit pathologies related to DNA-free and NET-associated histones or selectively inhibit DNA-free histones. This is a novel and interesting paper that provides important information about the possible therapeutic value of these SPAs, which are small poly sulfated di- or trisaccharides. However, several issues that were not well addressed in the paper should be considered by the authors, as follows:

1- The authors state in the Discussion that “Phase I clinical trials in healthy volunteers have also shown that mCBS is very well tolerated in humans at doses 20-fold higher than the maximum heparin dose administered to patients.” Given that the current paper appears to be meant to describe the discovery of such SPAs as mCBS, it is confusing to hear that it is already in clinical trials. Moreover, this sentence was not referenced by the authors. Accordingly, a better description of this point is important, as it could impact the novelty of this manuscript.

This manuscript is based on 10 years’ work, which was aimed at demonstrating that SPAs represent a new class of therapeutic that is able to neutralize pathologies mediated by extracellular histones, both free and NET-associated. The candidate SPAs, namely CBS, mCBS and MTS, were identified relatively early in this study and mCBS eventually selected for preclinical development by our commercial partner, Sirtex Medical Ltd, approximately 5 years ago. Thus, we have extensive preclinical data regarding the pharmacology and toxicology of mCBS as well as its safety in a Phase I clinical trial. This information has not been published previously and is submitted with this manuscript as a ‘Supplementary Information – Properties of mCBS.Na’ file, which is appropriately referred to in the revised manuscript.

2- Related to the comment above, the authors also state in the Discussion that “...mCBS was well tolerated in rats and dogs when continuously infused at 12.5 mg kg⁻¹ h⁻¹ for 14 days (unpublished data), which is 100 to 250-fold higher than the maximum dose of heparin recommended for human use.” This unpublished data needs to be presented in the current paper. In fact, this statement brings up many other things that need to be addressed by the current paper, such as a description of pharmacokinetics and dynamics, i.e. how long-lived are these small SPAs when injected, as in the current study, what is their clearance rate, how does this related temporally to their beneficial effects?

All of the requested information is available in the ‘Supplementary Information-Properties of mCBS.Na’ file.

3- It was important for the authors to compare their SPAs to heparin, in terms of efficacy in the pathology models, as well as in terms of untoward effects. One important effect I wondered about, even though acute administration of the SPAs did not increase platelet aggregation, is the pro-thrombin or clotting times after acute, vs chronic administration?

We agree that heparin would be an ideal comparator for the *in vivo* efficacy of our SPAs in the pathology models. Unfortunately, however, we have found that heparin is highly lethal when delivered at doses >6.25 mg kg⁻¹ (Fig. 4a), most animals receiving higher heparin doses dying within 4 h of heparin injection with evidence of internal hemorrhage. This result is not surprising as the mice receiving the higher heparin doses would be massively anti-coagulated. Nevertheless, based on a limited number of surviving mice at the higher doses we did observe that heparin gave similar levels of protection from histone-mediated tissue injury as mCBS at all 3 doses administered (Fig. 4b).

Furthermore, the anticoagulant activity of mCBS versus heparin is thoroughly examined and presented in the ‘Supplementary Information - Properties of mCBS.Na’ file, including the effects of acute versus chronic administration of mCBS on clotting time in healthy volunteers.

4- In terms of the cardiac IRI experiments shown in Fig. 6c the authors use IZ terminology to mean ischemic zone, but in IRI nomenclature “I” usually refers to infarct zone. A better terminology would be area at risk (AAR). And the authors used the term MVO for microvascular obstruction, and separately show infarct territory. It’s fine to show MVO but description of the infarct size should be just that, and is should be calculated as Infarct Area as a % of the AAR. MVO is a separate determination, and interesting to note, but not the same as infarct size. Furthermore, the authors should show visual examples of a CBS or MTS treated animal heart in addition to the saline treated heart they show.

The Figure 6c labeling has been corrected as suggested.

The images of the stained hearts were provided to us as an example of the scoring method used by an independent radiologist who analyzed the hearts. These images were included with the intention of providing greater clarity around the scoring method only, and since images for this particular experiment (performed 10 years ago) could not be retrieved, we have removed these images from Figure 6c.

5- Again, related to Fig. 6c, I’m not sure how the CBS vs MTS data shown here, where CBS was more effective than MTS and reducing infarct size, relates to Supplementary Fig. 2, where the right side of the figure shows MTS to be more potent or powerful than CBS in the ischemia models?

That was an error – thanks for spotting it! Ischemia was incorrectly labelled in the figure and incorrectly referred to in the Figure legend. This error has been corrected. The figure has also been modified to include our recent mechanistic insights into the mode of action of mCBS/CBS and MTS.

6- Also, What are the sera concentration of histones in the damage models and how are they affected by the supplementation of SPAs?

To answer this question a new method for detecting free histones in blood was developed. This involves exposing plasma to heparin-coupled beads, free histones binding with high affinity to such beads. The washed beads were then boiled in SDS-PAGE sample buffer, eluted proteins run on a 4%-20% gradient gel and protein bands detected by staining with Sypro Ruby. There are no plasma protein bands in the region of the gel in which histones migrate (see new Supplementary Fig. 5a), but plasma from i.v. histone (50 mg kg⁻¹) injected mice contained considerable amounts of free histones 1-2 min post histone injection, i.e., ~150 µg ml⁻¹ based on histone standards run in parallel tracks on the same gel. A major advantage of this procedure is that all histone isoforms can be easily detected simultaneously rather than only one at a time by Western blotting. We have not applied this technique yet to damage models but have used it to measure dramatically increased levels of histones in plasma from some sepsis patients.

When mCBS and MTS were injected i.p. into mice 10 min prior to i.v. injection of histones and plasma samples collected 1-2 min later, MTS produced a substantial and highly significant decline in histone levels (~80%) whereas mCBS had a smaller, but not statistically significant, effect (see new Supplementary Fig. 5b,c). This result is consistent with other data described in this manuscript that show that both *in vitro* and *in vivo* MTS is a consistently more effective inhibitor of free histones than mCBS. Importantly, these data also demonstrate that SPAs can rapidly interact with histones *in vivo* and remove them from the circulation.

The new experiments have resulted in a new Supplementary Fig. 5 and in new text associated with this new Figure (Supplementary File, page 5) and in the Methods section (Manuscript File, page 37, paragraph 2).

7- The authors should demonstrate in vivo evidence for the SPAs binding to histones if this is the presumed mechanism of protection and the kinetics of this binding, because SPAs were injected 5-minutes before reperfusion in cardiac IRI model and sacrificed only 30-minutes later implying rapid effect.

In vivo evidence of SPAs rapidly binding to histones is discussed in 6 above.

8- The authors should describe the rationale for such an acute cardiac IRI model (i.e. 30-min occlusion; 30-min reperfusion). Typically, in the literature the most acute timeframe of examining reperfusion injury where NETs was the focus is 3h, a time where the infarct is believed to be mostly due to intracellular ROS generation and calcium dysregulation in cardiac myocytes.

In the cardiac IRI model rats were subjected to 30 min rather than 2-3 h reperfusion because we found reperfusion injury was clearly evident at 30 min, with negligible mortality of reperfused rats, whereas mortality was a significant issue during 2-3 h reperfusion, an effect that potentially could bias results. This point is raised in the text of the Methods section in the revised manuscript (Manuscript File, page 41, paragraph 1, lines 11-14).

9- The authors should measure plasma cTnI levels upon cardiac IRI with and without SPA administration to delineate between protection from necrosis or protection that is secondary in action, as the authors propose.

Measuring plasma cTnI levels in cardiac IRI, with and without SPAs, is certainly worth investigating. However, such a study is beyond the scope of the present investigation, which is aimed at providing preliminary evidence that SPAs are active against a range of histone-mediated pathologies.

10- The authors should assess the effects of SPAs after more chronic cardiac IRI models, such as the 45 days post IRI injury in some of the authors' references, a time when functional benefit can be examined.

Our use of the cardiac IRI model in this study was to investigate whether CBS could affect the infarct territory in the area at risk, and efficacy was demonstrated (Fig. 6c) alongside a range of other models. It would be interesting to test the SPAs in more chronic cardiac IRI models but such studies are beyond the scope of this present study.

Reviewers' Comments:

Reviewer #1:

Remarks to the Author:

The authors have performed an extensive revision and I only have one minor remark left; in the new Figure S2, right panel, it seems that cytotoxicity for the two furthest right data points exceeds 100% - how is that possible?

Reviewer #2:

Remarks to the Author:

Overall, the authors have reasonably addressed the reviewers' comments. I have a few points to consider:

1. At several points in the paper, they reference "highly significant" (for example pg. 19). Results are significant or they are not significant. As such, the qualifier "highly" should be removed, as it does not add to the weight of the statement.
2. It is hard to see the staining in the lung of HIS+PBS in Figure 5b. Presumably, a high resolution figure will have better resolution. Otherwise, it is more fair to say kidney and liver.
3. I disagree with their assertion that the "best" way to compare molecules is through use of a gravimetric x-axis. Expression of the axis as equivalent moles of sulfate - for example - would be a more fair comparison. To say on page 5 that "their saccharides are as effective as unfractionated heparin" is not really true. Rather, based on the extensive pre-clinical (and as I understand it, clinical) analysis, it is more fair to say that the SPAs have high activity with a seemingly much better safety profile.
4. MINOR: The naming in the x-axis legend in Figure 4c should be reversed to be consistent with the rest of the Figure, i.e. "CBS+HIS" rather than "HIS+CBS"

Reviewer #3:

None

Reviewer #4:

Remarks to the Author:

The authors have done an admirable job of addressing most of my comments by including additional data in the supplement and by adjusting several of the figures in the main body of the manuscript.

Response to Reviewer's Comments

Accepted Manuscript NCOMMS-19-40206-T

Reviewer #1 (Remarks to the Author):

The authors have performed an extensive revision and I only have one minor remark left; in the new Figure S2, right panel, it seems that cytotoxicity for the two furthest right data points exceeds 100% - how is that possible?

In the new Supplementary Fig.2 the y-axis values can be >100% as they represent the ability of different concentrations of SPAs to inhibit histone cytotoxicity, the 100% control being histone cytotoxicity in the absence of inhibitors. Thus, unless histones alone kill 100% of cells, cytotoxicity values >100% can easily occur, in this case particularly when low concentrations of histone inhibitors (SPAs) are present.

Reviewer #2 (Remarks to the Author):

Overall, the authors have reasonably addressed the reviewers' comments. I have a few points to consider:

1. At several points in the paper, they reference "highly significant" (for example pg. 19). Results are significant or they are not significant. As such, the qualifier "highly" should be removed, as it does not add to the weight of the statement.

"Highly" has been removed from "highly significant" as requested by the reviewer.

2. It is hard to see the staining in the lung of HIS+PBS in Figure 5b. Presumably, a high resolution figure will have better resolution. Otherwise, it is more fair to say kidney and liver.

The cells that stain in the lung seem to be less fluorescent than those in liver or kidney (Fig 5b), but they can be easily seen when enlarged on a computer screen. In fact, when the number of PI+ cells was determined in lung and liver they were found to be almost identical (Fig. 5c).

3. I disagree with their assertion that the "best" way to compare molecules is through use of a gravimetric x-axis. Expression of the axis as equivalent moles of sulfate - for example - would be a more fair comparison. To say on page 5 that "their saccharides are as effective as unfractionated heparin" is not really true. Rather, based on the extensive pre-clinical (and as I understand it, clinical) analysis, it is more fair to say that the SPAs have high activity with a seemingly much better safety profile.

Unfortunately, this is an issue on which I differ from the reviewer. However, to ensure that readers of our manuscript understand that IC50 values in Fig. 1a are weight rather than molar based, I have modified the second last sentence of page 5 to read:

'Collectively, these data indicate that, **on a weight basis**, a highly sulfated hexose disaccharide is the minimum structure required for a SPA to inhibit histone-mediated cytotoxicity as effectively as heparin.'

4. MINOR: The naming in the x-axis legend in Figure 4c should be reversed to be consistent with the rest of the Figure, i.e. "CBS+HIS" rather than "HIS+CBS"

The naming of the x-axis of Figure 4c has been reversed as requested.

Reviewer #4 (Remarks to the Author):

The authors have done an admirable job of addressing most of my comments by including additional data in the supplement and by adjusting several of the figures in the main body of the manuscript.

No response required.